# TREM2 regulates purinergic receptor-mediated calcium signaling and motility in human iPSC-derived microglia

Amit Jairaman[1†], Amanda McQuade[2,3,4,5†‡], Alberto Granzotto[2,6,7], You Jung Kang[8], Jean Paul Chadarevian[2], Sunil Gandhi[2], Ian Parker[1,2], Ian Smith[2], Hansang Cho[9], Stefano L Sensi[6,7], Shivashankar Othy[1,10], Mathew Blurton-Jones[2,3,4,10*], Michael D Cahalan[1,10*]

[1]Department of Physiology and Biophysics, University of California, Irvine, Irvine, United States; [2]Department of Neurobiology and Behavior, University of California, Irvine, Irvine, United States; [3]Sue and Bill Gross Stem Cell Research Center, University of California, Irvine, Irvine, United States; [4]UCI Institute for Memory Impairments and Neurological Disorders, University of California, Irvine, United States; [5]Institute for Neurodegenerative Diseases, University of California, San Francisco, San Francisco, United States; [6]Center for Advanced Sciences and Technology (CAST), University "G. d'Annunzio" of Chieti-Pescara, Chieti, Italy; [7]Department of Neuroscience, Imaging and Clinical Sciences, University G d'Annunzio of Chieti-Pescara, Chieti, Italy; [8]Department of Mechanical Engineering and Engineering Science, University of North Carolina, Charlotte, United States; [9]Institute of Quantum Biophysics, Department of Biophysics, Dept of Intelligent Precision Healthcare Convergence, Sungkyunkwan University, Gyeonggi-do, Republic of Korea; [10]Institute for Immunology, University of California, Irvine, Irvine, United States

*For correspondence:
mblurton@uci.edu (MB-J);
mcahalan@uci.edu (MDC)

†These authors contributed equally to this work

‡This author is a co-first author

**Abstract** The membrane protein TREM2 (Triggering Receptor Expressed on Myeloid cells 2) regulates key microglial functions including phagocytosis and chemotaxis. Loss-of-function variants of TREM2 are associated with increased risk of Alzheimer's disease (AD). Because abnormalities in $Ca^{2+}$ signaling have been observed in several AD models, we investigated TREM2 regulation of $Ca^{2+}$ signaling in human induced pluripotent stem cell-derived microglia (iPSC-microglia) with genetic deletion of TREM2. We found that iPSC-microglia lacking TREM2 (TREM2 KO) show exaggerated $Ca^{2+}$ signals in response to purinergic agonists, such as ADP, that shape microglial injury responses. This ADP hypersensitivity, driven by increased expression of $P2Y_{12}$ and $P2Y_{13}$ receptors, results in greater release of $Ca^{2+}$ from the endoplasmic reticulum stores, which triggers sustained $Ca^{2+}$ influx through Orai channels and alters cell motility in TREM2 KO microglia. Using iPSC-microglia expressing the genetically encoded $Ca^{2+}$ probe, Salsa6f, we found that cytosolic $Ca^{2+}$ tunes motility to a greater extent in TREM2 KO microglia. Despite showing greater overall displacement, TREM2 KO microglia exhibit reduced directional chemotaxis along ADP gradients. Accordingly, the chemotactic defect in TREM2 KO microglia was rescued by reducing cytosolic $Ca^{2+}$ using a $P2Y_{12}$ receptor antagonist. Our results show that loss of TREM2 confers a defect in microglial $Ca^{2+}$ response to purinergic signals, suggesting a window of $Ca^{2+}$ signaling for optimal microglial motility.

## Editor's evaluation

Overall, this is a significant advance in the field of microglial regulation by calcium signaling.

## Introduction

As the primary immune cells of the central nervous system, microglia survey their local environment to maintain homeostasis and respond to local brain injury or abnormal neuronal activity. Microglia are strongly implicated in several neurodevelopmental and neurodegenerative diseases (*Andersen et al., 2021*; *Crotti et al., 2014*; *Fahira et al., 2019*; *Jansen et al., 2019*; *McQuade and Blurton-Jones, 2019*; *Pimenova et al., 2021*; *Tan et al., 2013*), warranting further study of human microglial dynamics. Purinergic metabolites (ATP, ADP, UTP, UDP) in the brain constitute key signals driving microglial activation and chemotaxis, and are detected by microglial cells over concentrations ranging from hundreds of nM to μM (*Davalos et al., 2005*; *De Simone et al., 2010*; *Honda et al., 2001*; *Koizumi et al., 2007*; *Haynes et al., 2006*; *Yegutkin, 2008*). ATP released from both homeostatic and damaged cells is hydrolyzed locally by nucleosidases such as the ectonucleotidase NTPDase1 (CD39) or pyrophosphatase NPP1 to produce ADP (*Dissing-Olesen et al., 2014*; *Madry and Attwell, 2015*; *Zhang et al., 2014*). ADP is then detected by P2Y purinergic receptors on microglia, causing $IP_3$-dependent $Ca^{2+}$ release from the endoplasmic reticulum (ER) lumen. $Ca^{2+}$ depletion from the ER in turn activates ER STIM1 proteins to translocate proximally to puncta where closely apposed plasma membrane (PM) Orai1 channels are activated. This mechanism underlies store-operated $Ca^{2+}$ entry (SOCE) in many cell types (*Prakriya and Lewis, 2015*), including microglia (*McLarnon, 2020*; *Mizuma et al., 2019*; *Gilbert et al., 2016*).

Purinergic signaling is central to microglial communication with other brain cell types and has been negatively correlated with the onset of disease-associated microglia (DAM) transcriptional states (*Hasselmann et al., 2019*; *Keren-Shaul et al., 2017*; *Krasemann et al., 2017*; *Olah et al., 2020*; *Sala Frigerio et al., 2019*). $P2Y_{12}$ and $P2Y_{13}$ receptors are highly expressed by microglia and are activated predominantly by ADP (*Zhang et al., 2014*; *Weisman et al., 2012*). $P2Y_{12}$ receptors are essential for microglial chemotaxis and have been implicated in the microglial response to cortical injury (*Haynes et al., 2006*; *Cserép et al., 2020*), NLRP3 inflammasome activation (*Suzuki et al., 2020*; *Wu et al., 2019*), neuronal hyperactivity and protection (*Cserép et al., 2020*; *Eyo et al., 2014*), and blood-brain barrier maintenance (*Lou et al., 2016*; *Bisht et al., 2021*). While purinergic receptors have been broadly identified as markers of microglial homeostasis (*Krasemann et al., 2017*; *Weisman et al., 2012*), mechanisms by which receptor expression may drive or maintain homeostatic microglial states remain incompletely understood.

Neuroinflammatory pathologies are often associated with altered $Ca^{2+}$ signaling (*Leissring et al., 2000*). Microglia, in particular, show altered $Ca^{2+}$ responses in mouse models of Alzheimer's disease (AD) by mechanisms that are not fully understood (*Brawek et al., 2014*; *Demuro et al., 2010*; *Mustaly-Kalimi et al., 2018*). $Ca^{2+}$ responses to purinergic metabolites have been extensively studied in cultured murine microglia, acute brain slices, and, more recently, in anesthetized mice (*Davalos et al., 2005*; *Honda et al., 2001*; *Brawek et al., 2014*; *Eichhoff et al., 2011*; *Irino et al., 2008*; *Milior et al., 2020*). However, our understanding of how specific patterns of $Ca^{2+}$ signals in microglia correlate with and tune downstream microglial responses such as cell motility or process extension remains incomplete. There is also a paucity of knowledge on how regulators of purinergic $Ca^{2+}$ signals in microglia might play a role in the dysregulation of $Ca^{2+}$ signaling associated with aging and neuroinflammation.

TREM2 encodes a cell surface receptor that binds a variety of ligands, including various lipids, apolipoprotein E (ApoE), and amyloid-β peptides. Upon ligand binding, TREM2 signals through its adaptor protein DAP12 to activate a host of downstream pathways (*Krasemann et al., 2017*; *Cheng-Hathaway et al., 2018*; *McQuade et al., 2020*; *Ulrich et al., 2014*). Loss of TREM2 function is thought to promote a more homeostatic-like state (*Krasemann et al., 2017*; *Andrews et al., 2020*; *Karch et al., 2012*). Indeed, microglia lacking TREM2 expression exhibit greatly diminished activation against disease pathology, correlating with increased risk of Alzheimer's disease (AD) (*Krasemann et al., 2017*; *McQuade et al., 2020*; *Cheng et al., 2018*). Purinergic receptor hyperexpression has been reported at the transcriptome level across multiple TREM2 loss of function models, including human patient mutations (*Hasselmann et al., 2019*; *Keren-Shaul et al., 2017*; *Krasemann et al., 2017*; *Sala Frigerio et al., 2019*; *McQuade et al., 2020*; *Gratuze et al., 2020*). For example, $P2Y_{12}$ receptor protein expression was found to be elevated in the cortical microglia of *Trem2*$^{-/-}$ mice and in a preclinical mouse model of AD (*Götzl et al., 2019*; *Griciuc et al., 2019*), although the mechanistic link between purinergic receptor expression and TREM2 function remains poorly understood.

We previously developed methods to generate human induced pluripotent stem cell-derived microglia (iPSC-microglia) (*Abud et al., 2017*; *McQuade et al., 2018*; *McQuade and Blurton-Jones, 2021*), which can be used to model human microglial behavior. While iPSC-microglia are proving increasingly useful to investigate neurodegenerative disorders (*McQuade et al., 2020*; *Andreone et al., 2020*; *Cosker et al., 2021*; *Konttinen et al., 2019*; *Piers et al., 2020*; *You et al., 2022*), $Ca^{2+}$ signaling has not yet been extensively profiled in these models. In this study, we compared purinergic $Ca^{2+}$ signaling and motility characteristics in wild type (WT) and TREM2 knockout (KO) human iPSC-microglia, and examined the mechanisms that underlie enhanced purinergic $Ca^{2+}$ signaling in microglia lacking TREM2. We find that motility is differentially tuned by $Ca^{2+}$ in TREM2 KO cells with consequences for chemotaxis.

## Results

### Purinergic receptor $Ca^{2+}$ signaling is enhanced in TREM2 KO human iPSC-microglia

To determine if TREM2 plays a role in microglial $Ca^{2+}$ signaling, we compared cytosolic $Ca^{2+}$ responses to the purinergic agonist ADP in isogenic, CRISPR-modified wild type (WT) and TREM2 KO human iPSC-microglia. ADP stimulation induced a biphasic $Ca^{2+}$ response – a rapid initial peak followed by a secondary phase of sustained $Ca^{2+}$ elevation lasting several minutes, in line with previous observations in mouse microglia (*Michaelis et al., 2015*; *Visentin et al., 2006*). Both phases of the $Ca^{2+}$ response were significantly elevated in TREM2 KO microglia, raising the possibility that augmentation of the initial $Ca^{2+}$ response to ADP in TREM2 KO microglia may be coupled to a larger sustained component of $Ca^{2+}$ entry (*Figure 1A and B*). These results were corroborated in iPSC-derived microglia cell line expressing the genetically encoded $Ca^{2+}$ indicator Salsa6f (*Dong et al., 2017*; *Jairaman and Cahalan, 2021*; *Figure 1C and D*). The Salsa6f probe showed the expected increase in the GCaMP6f fluorescence in response to $Ca^{2+}$ elevation without any change in the tdTomato signal, and it did not perturb microglial activation and function (*Figure 1—figure supplement 1A–G*). TREM2 KO microglia also showed exaggerated $Ca^{2+}$ responses to the purinergic agonists ATP and UTP at similar low µM concentrations, although the secondary $Ca^{2+}$ elevations were not as long-lasting as with ADP (*Figure 1E and F*, *Figure 1—figure supplement 2*).

### Increased $P2Y_{12}$ and $P2Y_{13}$ receptor expression drives increased peak $Ca^{2+}$ in TREM2 KO microglia

Given the critical importance of ADP signaling in several aspects of microglial function, we investigated the mechanisms driving higher ADP-evoked $Ca^{2+}$ signals in TREM2 KO microglia by focusing on specific steps in the purinergic $Ca^{2+}$ signaling pathway (*Figure 2A*). The initial $Ca^{2+}$ response to P2Y receptor engagement results from G protein-coupled phospholipase C activation and $IP_3$-mediated ER $Ca^{2+}$ store release. To test this, we treated cells with ADP in $Ca^{2+}$-free solution buffered with the $Ca^{2+}$ chelator EGTA to isolate $Ca^{2+}$ signals from store release and eliminate $Ca^{2+}$ influx across the PM. Both WT and TREM2 KO cells exhibited a single $Ca^{2+}$ peak, with TREM2 KO cells showing significantly higher peak $Ca^{2+}$ response to ADP (*Figure 2B*, *Figure 2—figure supplement 1A and B*). Moreover, the amplitude of the $Ca^{2+}$ peak was not significantly different in the presence or absence of external $Ca^{2+}$, strongly suggesting that it is driven primarily by release of $Ca^{2+}$ from intracellular stores even when external $Ca^{2+}$ is present (*Figure 2—figure supplement 1C*). Dose–response curves for the peak $Ca^{2+}$ response showed a steep leftward shift in TREM2 KO cells (*Figure 2C*). The $EC_{50}$ value for WT microglia was 650 nM, whereas TREM2 KO microglia reached their $EC_{50}$ by 15 nM. This stark difference was driven at least in part by a diminished percentage of WT cells responding to ADP at low µM doses (*Figure 2D*). However, limiting the analysis to cells that showed a $Ca^{2+}$ rise revealed that 'responding' TREM2 KO cells still exhibited higher $Ca^{2+}$ responses to ADP than 'responding' WT cells (*Figure 2E*). TREM2 KO microglia are thus significantly more sensitive to ADP than WT cells, which may be critical in sensing ADP and detecting ADP gradients.

RNA-sequencing revealed significantly increased transcripts for $P2Y_{12}$ and $P2Y_{13}$ receptors, the main P2Y receptor subtypes in microglia that bind ADP, in TREM2 KO microglia (*Abud et al., 2017*; *McQuade et al., 2018*; *Figure 2F*). In comparison, relative mRNA levels of common mediators of $Ca^{2+}$ signaling – including predominant isoforms of $IP_3$ receptors, SOCE mediators Orai and STIM proteins,

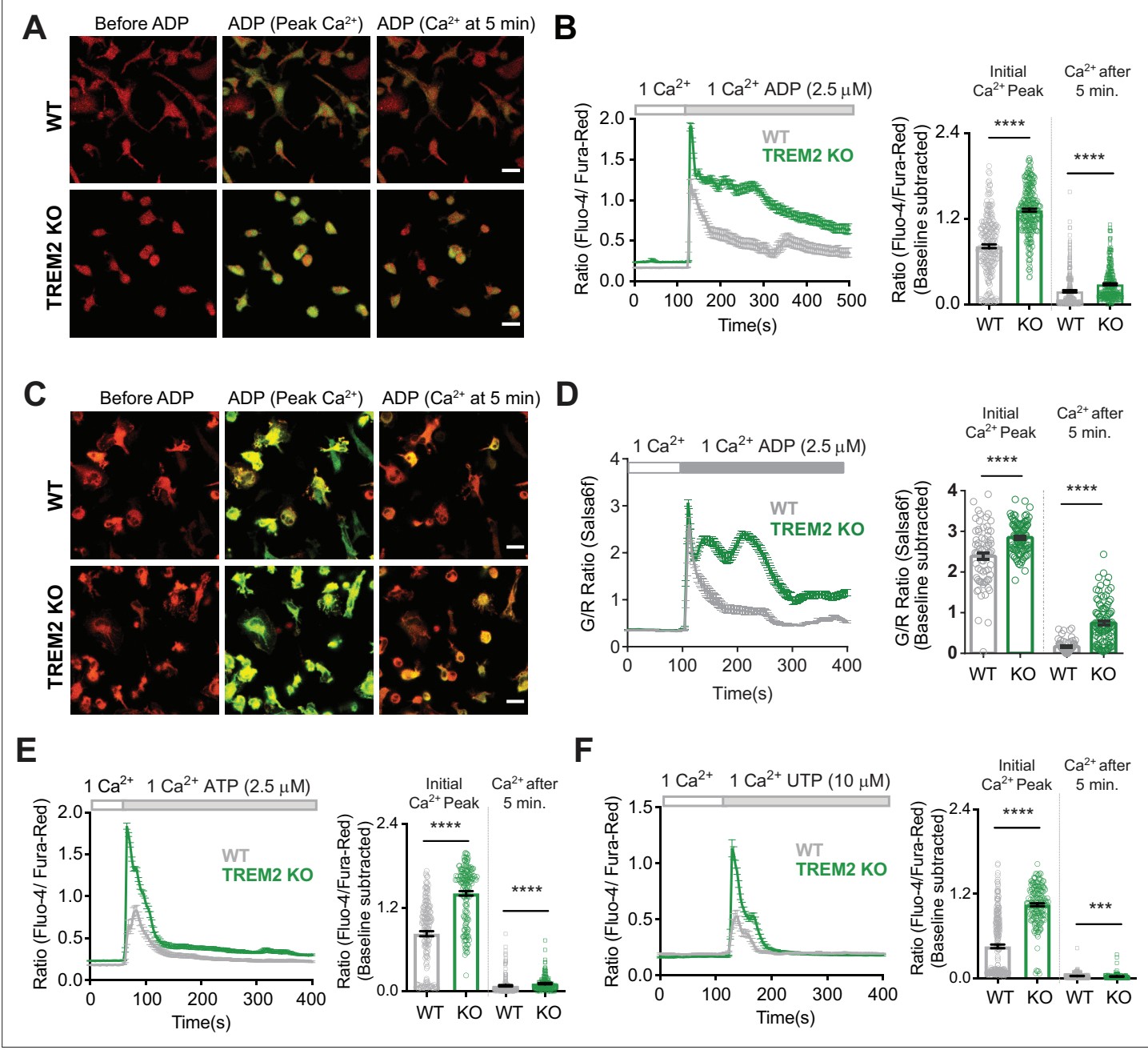

**Figure 1.** Microglia lacking TREM2 show exaggerated $Ca^{2+}$ responses to purinergic stimulation. (**A**) Representative red-green channel overlay images of wild type (WT) (top) and TREM2 knockout (KO) (bottom) induced pluripotent stem cell (iPSC)-microglia loaded with Fluo-4 (green) and Fura-red (red) showing resting cytosolic $Ca^{2+}$ before ADP, and $Ca^{2+}$ levels 15 s and 5 min after ADP addition. Scale bar = 20 μm. (**B**) Average traces (left panels) showing changes in cytosolic $Ca^{2+}$ in response to 2.5 μM ADP in 1 mM $Ca^{2+}$ buffer (n = 39–44 cells). Baseline-subtracted peak $Ca^{2+}$ response and cytosolic $Ca^{2+}$ levels 5 min after ADP shown on the right (n = 250–274 cells, five experiments, Mann–Whitney test). (**C, D**) Cytosolic $Ca^{2+}$ response to ADP as in (**A**) and (**B**) but in iPSC-microglia expressing the GCaMP6f-tdTomato fusion $Ca^{2+}$ probe Salsa6f (n = 41–53 cells, two independent experiments, Mann–Whitney test). Images in (**C**) are overlay of GCaMP6f (green) and tdTomato (red) channel images. Scale bar = 20 μm. (**E**) $Ca^{2+}$ responses to 2.5 μM ATP in WT and TREM2 KO iPSC-microglia. Average traces (left panel, n = 63–71 cells) and bar graph summary of peak cytosolic $Ca^{2+}$ and $Ca^{2+}$ after 5 min (right panel, 165–179 cells, three experiments, Mann–Whitney test). (**F**) $Ca^{2+}$ responses to 10 μM UTP. Average traces (45–55 cells) and summary of peak cytosolic $Ca^{2+}$ and $Ca^{2+}$ after 5 min (175–269 cells, three experiments, Mann–Whitney test). Data shown as mean ± SEM for traces and bar graphs. p-Values indicated by *** for p<0.001, ****p<0.0001.

The online version of this article includes the following source data and figure supplement(s) for figure 1:

**Source data 1.** Microglia lacking TREM2 show exaggerated $Ca^{2+}$ responses to purinergic stimulation.

*Figure 1 continued on next page*

*Figure 1 continued*

**Figure supplement 1.** Validation of Salsa6f transgenic induced pluripotent stem cell (iPSC)-microglia.

**Figure supplement 2.** Comparison of cytosolic $Ca^{2+}$ signal over time triggered by various purinergic agonists.

and SERCA and PMCA $Ca^{2+}$ pumps – were either similar or modestly reduced in TREM2 KO in comparison with WT iPSC-microglia (*Figure 2—figure supplement 1D and E*). We therefore considered the possibility that signal amplification in microglia lacking TREM2 results primarily from increased expression of $P2Y_{12}$ and $P2Y_{13}$ receptors. Consistent with this, expression of $P2Y_{12}$ receptors in the PM was significantly increased in TREM2 KO cells (*Figure 2G*). Furthermore, $Ca^{2+}$ responses to ADP in $Ca^{2+}$-free medium were completely abolished following treatment with a combination of $P2Y_{12}$ and $P2Y_{13}$ receptor antagonists (PSB 0739 and MRS 2211, respectively) in both WT and TREM2 KO microglia (*Figure 2H*). Treatment of cells with $P2Y_{12}$ and $P2Y_{13}$ receptor antagonists separately produced partial inhibition of peak ADP-mediated $Ca^{2+}$ signals, implicating involvement of both receptor subtypes (*Figure 2—figure supplement 1F and G*). In summary, deletion of TREM2 results in a larger cytosolic $Ca^{2+}$ peak in response to ADP due to increased expression of $P2Y_{12}$ and $P2Y_{13}$ receptors.

## SOCE through Orai channels mediates the sustained phase of ADP-evoked $Ca^{2+}$ elevation

To probe the basis for the increased sustained component of ADP-evoked $Ca^{2+}$ signal in TREM2 KO microglia, we examined SOCE using pharmacological and genetic approaches. Synta66, a reasonably specific inhibitor of Orai channels, significantly reduced the rate of SOCE following $Ca^{2+}$ readdition after ER store depletion by the sarco-endoplasmic reticulum $Ca^{2+}$ ATPase (SERCA pump) inhibitor, thapsigargin (TG), in both WT and TREM2 KO microglia (*Figure 3A*, *Figure 3—figure supplement 1A*). Using a similar $Ca^{2+}$ readdition protocol with ADP, we found significant inhibition of ADP-induced SOCE by Synta66 in both WT and TREM2 KO cells (*Figure 3B*, *Figure 3—figure supplement 1B*). The ADP-evoked sustained $Ca^{2+}$ phase in TREM2 KO iPSC-microglia was also blocked by less specific Orai channel inhibitors, $Gd^{3+}$ and 2-APB (*Figure 3—figure supplement 1C and D*). To further confirm the specific role of Orai1 channels in mediating SOCE, we generated an Orai1 CRISPR-knockout iPSC line. Deletion of Orai1 abrogated SOCE and significantly reduced the sustained $Ca^{2+}$ response to ADP (*Figure 3—figure supplement 1E and F*). These results confirm that Orai1 plays an important role in mediating SOCE and ADP-evoked $Ca^{2+}$ signals in iPSC-microglia.

To determine if SOCE is increased in TREM2 KO microglia and contributing to the higher sustained $Ca^{2+}$ response to ADP, we compared the rate of store-operated $Ca^{2+}$ influx after store depletion with TG and found that both the rate and amplitude of SOCE were modestly reduced in TREM2 KO cells (*Figure 3C*). In keeping with this, RNA-sequencing revealed a modest reduction in STIM1 mRNA expression in TREM2 KO cells, although Orai1 mRNA was similar in WT and TREM2 KO microglia (*Figure 2—figure supplement 1C and D*). We further conclude that the elevated secondary phase of ADP-driven $Ca^{2+}$ signals in TREM2 KO microglia is not primarily due to the differences in the expression of STIM and Orai.

## ADP depletes ER $Ca^{2+}$ stores to a greater extent in TREM2 KO microglia, leading to greater SOCE activation

We hypothesized that the exaggerated secondary $Ca^{2+}$ phase in response to ADP in TREM2 KO microglia may be driven by increased ER $Ca^{2+}$ store release, leading to greater SOCE activation. Consistent with this possibility, peak cytosolic $Ca^{2+}$ in response to partial store depletion with ADP and after $Ca^{2+}$ readdition was elevated in TREM2 KO microglia (*Figure 3D*). To examine if the higher magnitude of SOCE in TREM2 KO cells is due to depletion of ER $Ca^{2+}$ stores by ADP, we sequentially treated cells with ADP followed by ionomycin to completely release stores in $Ca^{2+}$-free buffer. While TREM2 KO cells showed greater peak $Ca^{2+}$ with ADP as expected, the ionomycin $Ca^{2+}$ peak – which reflects the residual ER $Ca^{2+}$ pool – was significantly reduced, indicating that ADP depletes ER $Ca^{2+}$ stores to a greater extent in TREM2 KO cells (*Figure 3E*). Similar results were obtained when residual ER store content was depleted using TG instead of ionomycin (*Figure 3—figure supplement 2A and B*). We plotted cytosolic $Ca^{2+}$ levels 5 min after addition of varying doses of ADP to indicate the degree of SOCE as a function of the initial peak $Ca^{2+}$, a readout of ER store release (*Figure 3—figure*

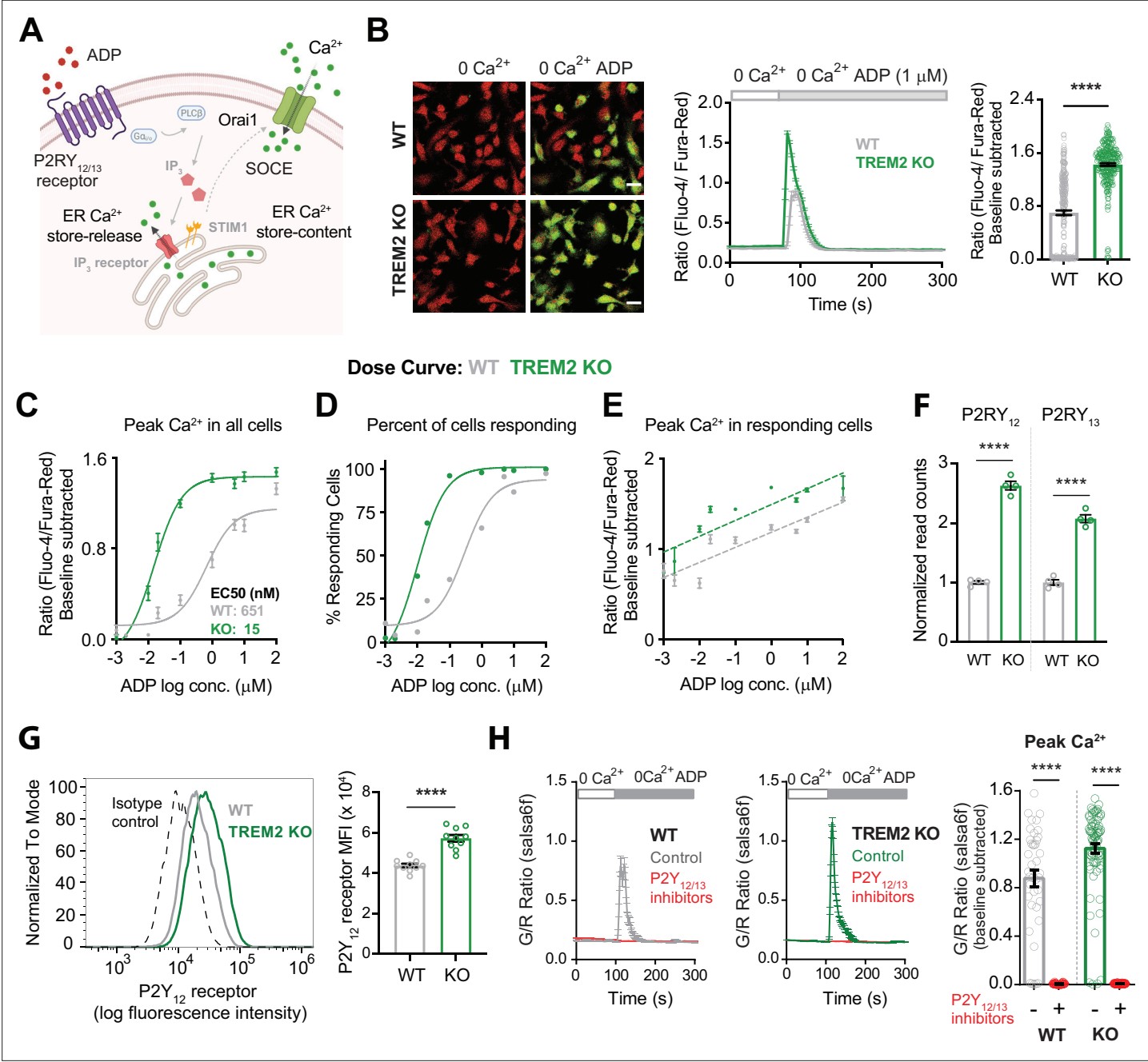

**Figure 2.** Higher sensitivity of TREM2 knockout (KO) microglia to ADP is driven by increased purinergic receptor expression. (**A**) Schematic highlighting key downstream $Ca^{2+}$ signaling events triggered by ADP. Cytosolic $Ca^{2+}$ response to ADP is determined by functional expression and activity of $P2Y_{12}$ and $P2Y_{13}$ receptors, $IP_3$ receptors, endoplasmic reticulum (ER) store $Ca^{2+}$ content, and store-operated $Ca^{2+}$ entry (SOCE) regulated by STIM and Orai proteins. (**B**) Representative images (left panel) showing overlay of Fluo-4 (green) and Fura-red (red) channels in wild type (WT) (top) and TREM2 KO (bottom) induced pluripotent stem cell (iPSC)-microglia before and peak $Ca^{2+}$ response after ADP addition in $Ca^{2+}$-free buffer. Scale bar = 20 μm. Average trace showing $Ca^{2+}$ response to ADP in $Ca^{2+}$-free buffer (middle panel, 64–83 cells). Quantification of peak signal (right panel, n = 264–289 cells, four experiments, Mann–Whitney test). (**C–E**) Dose–response curves showing baseline-subtracted peak $Ca^{2+}$ responses to ADP in $Ca^{2+}$-free buffer (**C**), percent of 'responding' cells (**D**), and peak $Ca^{2+}$ responses only in 'responding' cells (**E**). N = 84–474 WT cells and 70–468 TREM2 KO cells, 2–5 experiments. (**F**) RNA normalized read counts of $P2Y_{12}$ and $P2Y_{13}$ receptor expression from bulk RNA-sequencing of WT and TREM2 KO iPSC-microglia (n = 4, adjusted p-values from DESeq2). (**G**) Representative histogram (left panel) showing plasma membrane (PM) expression of $P2Y_{12}$ receptor in WT and TREM2 KO microglia. Cells were stained with BV421-labeled anti-human $P2Y_{12}$ receptor antibody. Isotype control is shown as dashed line. Right panel shows summary of median fluorescence intensity (MFI) of $P2Y_{12}$ receptor-labeled cells (n = 10 samples each, Student's *t*-test). (**H**) $Ca^{2+}$ traces (left panel) showing response to 1 μM ADP in $Ca^{2+}$-free buffer after 30 min pretreatment with a combination of $P2Y_{12}$ receptor antagonist PSB 0739 (10 μM)

*Figure 2 continued on next page*

*Figure 2 continued*

and P2Y$_{13}$ receptor antagonist MRS 2211 (10 µM). Summary of the peak Ca$^{2+}$ response (right panel, n = 40–79 cells, two experiments, Mann–Whitney test). Data are mean ± SEM. p-Values indicated by ****p<0.0001.

The online version of this article includes the following source data and figure supplement(s) for figure 2:

**Source data 1.** Higher sensitivity of TREM2 knockout (KO) microglia to ADP is driven by increased purinergic receptor expression.

**Figure supplement 1.** Role of P2Y$_{12}$ and P2Y$_{13}$ receptors in ADP-mediated augmentation of store release in TREM2 knockout (KO) microglia.

*supplement 2C and D*). Both WT and TREM2 KO microglia showed similar linear relationships between SOCE and store release, further suggesting that SOCE is activated by similar mechanisms in the two cell lines, but is recruited to a greater extent in TREM2 KO cells due to increased ER store release. We also note that increased sustained Ca$^{2+}$ in TREM2 KO cells is unlikely to be due to differences in Ca$^{2+}$ pump activity based on similar Ca$^{2+}$ clearance rates (*Figure 3—figure supplement 2E and F*), consistent with comparable transcriptomic expression of major SERCA and PM Ca$^{2+}$ ATPase (PMCA) isoforms in WT and TREM2 KO cells (*Figure 2—figure supplement 1C and D*).

Finally, quantification of cumulative cytosolic Ca$^{2+}$ increases after maximally depleting ER stores with ionomycin alone suggested that overall ER store content is not altered in microglia lacking TREM2 (*Figure 3F*). Comparison of Ca$^{2+}$ responses to IP$_3$ uncaging also ruled out major differences in the pool of functional IP$_3$ receptors between WT and TREM2 KO cells (*Figure 3G*), as further substantiated by similar transcriptomic expression of IP$_3$ receptor type 2 (the major IP$_3$R subtype expressed in iPSC-microglia) in WT and TREM2 KO cells (*Figure 2—figure supplement 1C and D*; *McQuade et al., 2020*; *Abud et al., 2017*). In summary, deletion of TREM2 in iPSC-derived microglia leads to upregulation of P2Y$_{12}$ and P2Y$_{13}$ receptors and renders the cells hypersensitive to ADP signaling, consequently leading to greater IP$_3$-mediated ER store depletion and increased coupling to SOCE in response to purinergic metabolites.

## ADP potentiates cell motility and process extension in human WT iPSC-microglia

ADP is a potent chemoattractant for microglia (*Honda et al., 2001*). Analogous to a previous study in fibroblasts (*Borges et al., 2021*), we found that ADP treatment alters cell motility and leads to increased rates of scratch wound closure in WT iPSC-microglia (*Figure 4A*). To investigate the cellular mechanism of accelerated wound closure, we used time-lapse imaging to track open-field microglial cell motility (*Figure 4B*). Mean cell track speed and track displacement (defined as the overall change in position from the origin at a given time) were both increased after application of ADP. On the other hand, average track straightness, an indicator of how frequently cells change direction, was unaltered by ADP (*Figure 4C*). These data suggest that ADP-driven changes in motility in WT iPSC-microglia primarily arise from increases in microglial speed, and not altered turning behavior. ADP-dependent increases in speed were reversed in the presence of P2Y$_{12}$ (PSB 0739) and P2Y$_{13}$ (MRS 2211) receptor antagonists, confirming the role of these two purinergic receptors in ADP enhancement of microglial motility (*Figure 4D*). To determine if Ca$^{2+}$ influx regulates ADP-mediated increases in motility, we measured cell migration with ADP in Ca$^{2+}$-free medium and found that removing extracellular Ca$^{2+}$ significantly decreased cell speed, displacement, and track straightness, suggesting that sustained Ca$^{2+}$ signals are required for maximal increase in motility in response to ADP (*Figure 4E*).

In addition, some microglia responded to ADP by extending processes and altering their morphology rather than increasing motility (*Figure 4—figure supplement 1*). Microglia have been observed to extend processes in response to injury and purinergic stimulation in brain slices (*Davalos et al., 2005*; *Haynes et al., 2006*). Therefore, we compared process complexity before and 30 min after ADP exposure in WT microglia and observed significant increases in both the number of branches per process and total length of these processes (*Figure 4F*). Similar to effects on cell motility, ADP-mediated process extension was inhibited by P2Y$_{12}$ and P2Y$_{13}$ receptor antagonists (PSB 0739 and MRS 2211, respectively). Furthermore, even before process extension was activated with ADP, cells treated with P2Y antagonists showed significantly fewer and shorter processes, suggesting that baseline purinergic signaling may regulate resting microglial process dynamics. Altogether, these results demonstrate that activation of purinergic signaling through P2Y$_{12}$ and P2Y$_{13}$ receptors is required for ADP-driven microglial process extension and motility.

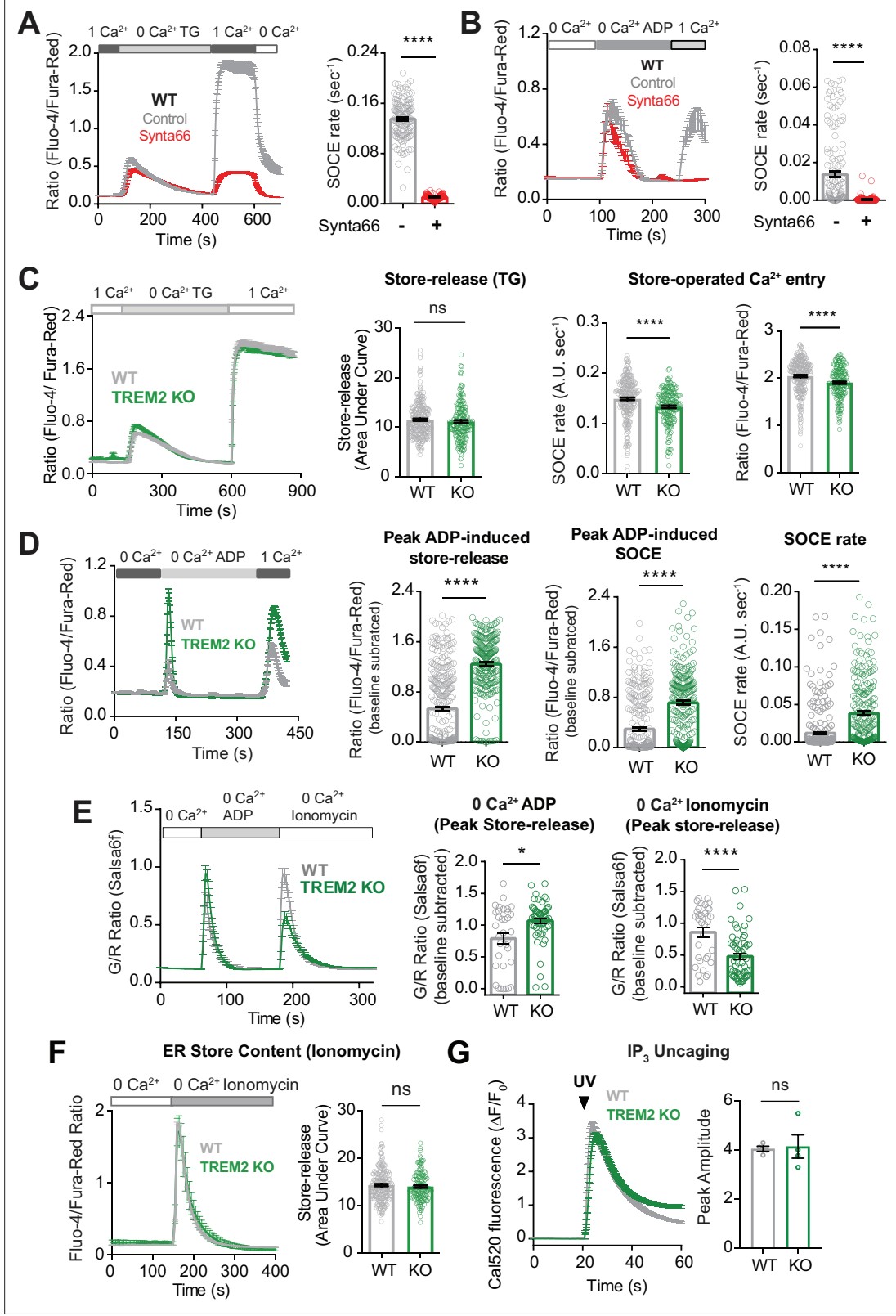

**Figure 3.** Regulation of ADP-evoked store-operated Ca²⁺ entry (SOCE) in wild type (WT) and TREM2 knockout (KO) microglia. (**A**) SOCE in WT microglia triggered with thapsigargin (TG, 2 µM) in Ca²⁺-free buffer followed by readdition of 1 mM Ca²⁺ in the absence (control, gray trace) or presence (red trace) of the Orai channel inhibitor Synta66 (n = 34–48 cells). Cells were pretreated with Synta66 (10 µM) for 30 min before imaging. Bar graph summary of the rate of Ca²⁺ influx (n = 80–137 cells, two experiments, Mann–Whitney test). (**B**) SOCE evoked by ADP (2.5 µM) in WT microglia (gray trace) using a

*Figure 3 continued on next page*

*Figure 3 continued*

similar Ca²⁺ addback protocol as in (**A**). Red trace shows the effect of Synta66 on ADP-evoked SOCE. Right panel shows bar graph summary of the rate of ADP-triggered Ca²⁺ influx after readdition of 1 mM Ca²⁺ (n = 148–155 cells, two experiments, Mann–Whitney test). (**C**) Comparison of SOCE evoked with TG (2 µM) in WT and TREM2 KO cells (n = 90–129 cells). Bar graph summaries of endoplasmic reticulum (ER) store release quantified as area under the curve, rate of SOCE, and peak SOCE (n = 187–266 cells, two experiments, Mann–Whitney test). (**D**) Traces showing ADP-evoked SOCE in WT and TREM2 KO microglia after depleting stores with 100 nM ADP in Ca²⁺-free buffer and readdition of 1 mM Ca²⁺ (left panel, n = 97–114 cells). Comparison of ADP-evoked cytosolic Ca²⁺ peak, peak SOCE and SOCE rate (right panel, n = 234–313 cells, three experiments, Mann–Whitney test). (**E**) Ionomycin pulse experiment to measure residual ER Ca²⁺ pool in cells after initial treatment with ADP. WT and TREM2 KO cells were pulsed sequentially with ADP first (200 nM) and subsequently treated with ionomycin (1 µM) to empty and measure the residual pool of ER Ca²⁺. Imaging was done entirely in Ca²⁺-free buffer to prevent Ca²⁺ influx across the plasma membrane (PM). Average trace (left panel), peak ADP Ca²⁺ response (middle panel), and peak ionomycin-induced Ca²⁺ response (right panel) (n = 38–60 cells, 3–4 experiments, Mann–Whitney test). (**F**) Average trace (left, 71–117 cells) and summary of ER store release after 2 µM ionomycin treatment in Ca²⁺-free buffer (right, 146–234 cells, two experiments; nd, nonsignificant p>0.05, Mann–Whitney test). (**G**) Same as (**H**) but in response to UV IP₃ uncaging (167–200 cells, ns, nonsignificant p>0.05, nonparametric t-test). Data shown as mean ± SEM for traces and bar graphs. Data are mean ± SEM. p-Values indicated by ns, nonsignificant, *p<0.05, and ****p<0.0001.

The online version of this article includes the following source data and figure supplement(s) for figure 3:

**Source data 1.** Regulation of ADP-evoked store-operated Ca²⁺ entry (SOCE) in wild type (WT) and TREM2 knockout (KO) microglia.

**Figure supplement 1.** Regulation of store-operated Ca²⁺ entry (SOCE) in induced pluripotent stem cell (iPSC)-microglia.

**Figure supplement 2.** ADP depletes endoplasmic reticulum (ER) Ca²⁺ stores to a greater extent in TREM2 knockout (KO) microglia.

## ADP-evoked changes in cell motility and process extension are enhanced in TREM2 KO microglia

To characterize differences in motility characteristics between WT and TREM2 KO microglia responding to ADP, we plotted mean squared displacement (MSD) vs. time and compared cell track overlays (flower plots), which showed that ADP enhances motility in TREM2 KO cells to a greater extent than in WT microglia (*Figure 5A and B*). Baseline motility characteristics in unstimulated cells, however, were similar in WT and TREM2 KO cells (*Figure 5—figure supplement 1A and B*). To further understand the basis of differences in ADP-induced motility between WT and TREM2 KO cells, we analyzed mean track speed, track displacement, and track straightness. Although mean track speeds were similar, TREM2 KO microglia showed greater displacement than WT cells (*Figure 5C and D*), raising the possibility that TREM2 KO cells may turn with lower frequency. Consistent with this, analysis of track straightness revealed that TREM2 KO microglia move farther from their origin for the same total distance traveled (*Figure 5E*). Vector autocorrelation, an analysis of directional persistence (*Gorelik and Gautreau, 2014*), further confirmed that WT cells turn more frequently than TREM2 KO microglia in response to ADP (*Figure 5—figure supplement 1C and D*). To assess if these differences in TREM2 KO cells require sustained Ca²⁺ influx, we analyzed microglial motility in response to ADP stimulation in the absence of extracellular Ca²⁺ (*Figure 5F–J*). MSD and cell track overlay plots showed that motility is constrained when Ca²⁺ is removed from the external bath in both WT and TREM2 KO cells (*Figure 5A and B vs. F and G*). In the absence of extracellular Ca²⁺, TREM2 KO microglia showed similar mean speed, displacement, and track straightness as WT cells (*Figure 5C–E vs. H–J*). We conclude that increases in microglial motility (mean speed, displacement, and straightness) require sustained Ca²⁺ influx and that deletion of TREM2 reduces microglial turning in response to ADP.

We next analyzed the effects of TREM2 deletion on process extension in microglia. Treatment with ADP induced a dramatic increase in the number of branches and length of processes extended in both WT and TREM2 KO microglia (*Figure 5K and L*). Comparison of the absolute number of branches and process length after ADP treatment, as well as the relative fold increase in these parameters from baseline, indicated that process extension is not affected in TREM2 KO microglia (*Figure 5K–M*, *Figure 5—figure supplement 2A and B*). We note that the greater fold change in process extension in TREM2 KO cells can be attributed to the reduced morphological complexity of these cells prior to stimulation. Finally, ADP stimulation in Ca²⁺-free medium did not induce process extension in WT cells, and only a modest increase in TREM2 KO cells (*Figure 5—figure supplement 2A and B vs. C and D*). Together, these results indicate that sustained Ca²⁺ entry across the PM is required for optimal microglial process extension in both WT and TREM2 KO microglia.

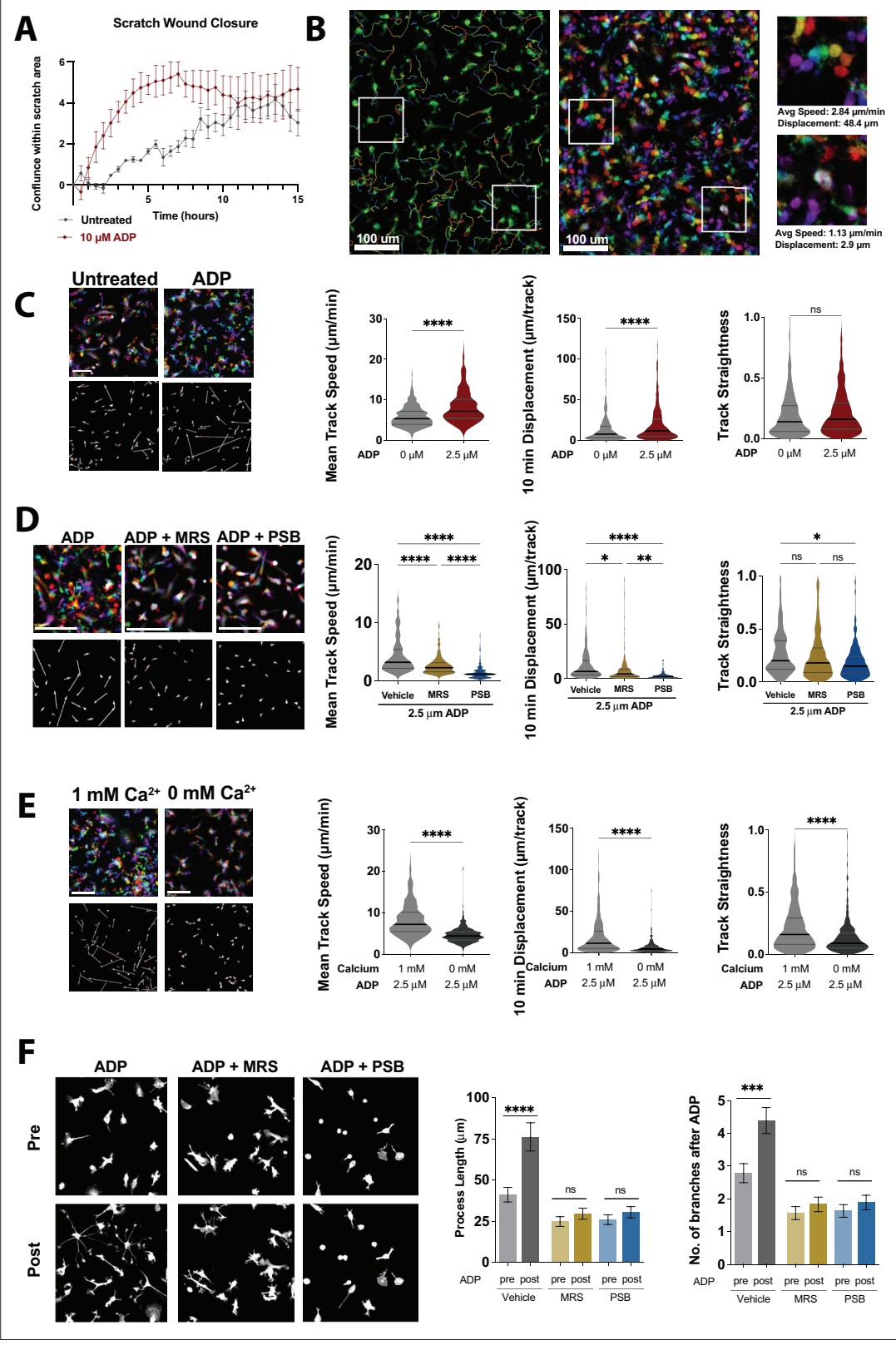

**Figure 4.** Nondirectional ADP exposure increases wild type (WT) microglial speed and process extension. (**A**) Average trace showing closure of scratch wound produced with IncuCyte S3 WoundMaker. Induced pluripotent stem cell (iPSC)-microglia imaged every 30 min after scratch wound with or without ADP stimulation (n = 4 wells; two images per well). (**B**) Representative image of WT iPSC-microglia motility 30 min after ADP exposure with cell

*Figure 4 continued on next page*

*Figure 4 continued*

tracks overlain (left). Pseudocolored images (center) across time: 0 min (red), 4 min (orange), 8 min (yellow), 12 min (green), 16 min (cyan), 20 min (blue), 24 min (purple), and 28 min (magenta). Scale bar = 100 µm. White boxes zoomed in at right to demonstrate motile (top) and nonmotile (bottom) cells. (**C**) Representative color images (top left) and displacement vectors (bottom left) of WT iPSC-microglia at baseline (no ADP, gray) and 30 min after 2.5 µM ADP treatment (red). Summary of mean speed (µm/min), Displacement over 10 min (µm/10 min) and track straightness (track length/track displacement) (414–602 cells, two experiments). (**D**) Representative images, displacement vectors, and quantification of WT iPSC-microglia motility for 20 min following ADP addition. Cells were pretreated with vehicle (gray), MRS 2211 (10 µM, gold), or PBS 0739 (10 µM, blue) (180–187 cells, two experiments). (**E**) Representative images, displacement vectors, and quantification of WT iPSC-microglia motility after ADP in 1 mM $Ca^{2+}$ (light gray) or $Ca^{2+}$-free buffer (dark gray) (401–602 cells, three experiments). (**F**) Representative images (left) and process extension (right) of iPSC-microglia (cytoplasmic GFP, gray) before or 30 min after ADP addition. Cells were pretreated with vehicle (gray), MRS 2211 (10 µM, gold), or PBS 0739 (10 µM, blue) (52–163 cells, 3–4 experiments). (**C–F**) One-way ANOVA with Tukey post hoc test. Data shown as mean ± SEM (**A, F**) and as violin plots with mean, 25th and 75th percentile (**C–E**). p-Values indicated by ns, nonsignificant, *p<0.05, **p<0.01, ***p<0.001, and ****p<0.0001.

The online version of this article includes the following source data and figure supplement(s) for figure 4:

**Source data 1.** Nondirectional ADP exposure increases wild type (WT) microglial speed and process extension.

**Figure supplement 1.** ADP-mediated process extension in wild type (WT) induced pluripotent stem cell (iPSC)-microglia.

## Cytosolic $Ca^{2+}$ levels tune motility in TREM2 KO iPSC-microglia

To further characterize the effects of sustained $Ca^{2+}$ signals on microglial motility, we used Salsa6f-expressing iPSC WT and TREM2 KO reporter lines to monitor cytosolic $Ca^{2+}$ and motility simultaneously in individual cells (*Figure 6—figure supplement 1*). To isolate the effects of sustained $Ca^{2+}$ elevations on microglia motility and eliminate any contribution from $Ca^{2+}$ independent signaling pathways, we used a protocol that relies on triggering SOCE and varying external $Ca^{2+}$ to maintain cytosolic $Ca^{2+}$ at 'low' or 'high' levels in the Salsa6f reporter line (*Figure 6A–C*), similar to our previous study in T lymphocytes (*Negulescu et al., 1996*). In WT cells, lowering extracellular $Ca^{2+}$ from 2 to 0.2 mM predictably decreased the G/R ratio but did not influence mean track speed, 10 min track displacement, or track straightness (*Figure 6C and D*, top). However, in TREM2 KO microglia, reducing $Ca^{2+}$ to a lower level significantly increased speed, displacement, and track straightness (*Figure 6C and D*, bottom). These data suggest that motility characteristics of TREM2 KO microglia are more sensitive to changes in cytoplasmic $Ca^{2+}$ levels than in WT cells. Similar results were obtained upon addition of ADP in this paradigm, suggesting that long-lasting $Ca^{2+}$ elevations may override effects of $Ca^{2+}$-independent ADP signaling on cell motility (*Figure 6—figure supplement 2A*).

To further analyze the $Ca^{2+}$ dependence of microglial motility, we plotted Salsa6f G/R $Ca^{2+}$ ratios for each individual cell at every time point against the instantaneous speeds of that cell (*Figure 6E*). These data revealed a stronger dependence of instantaneous speed on $Ca^{2+}$ levels in TREM2 KO microglia (*Figure 6F*). Furthermore, when stratifying cell speed arbitrarily as 'fast' (>10 µm/min) or 'slow' (<10 µm/min), we observe a marked reduction in the percentage of 'fast' cells when $Ca^{2+}$ levels are high in TREM2 KO microglia (*Figure 6G*). Interestingly, frame-to-frame cell displacement correlated with cytosolic $Ca^{2+}$ to the same degree in both WT and KO cells (*Figure 6—figure supplement 2B and C*). Together, TREM2 KO human microglia are more sensitive to tuning of motility by cytosolic $Ca^{2+}$ than WT cells.

## Chemotactic defects in TREM2 KO microglia are rescued by dampening purinergic receptor activity

To assess the physiological significance of TREM2 deletion on microglial motility over longer time scales, we performed a scratch wound assay. At baseline, both WT and TREM2 KO microglia migrated into the cell-free area at similar rates, consistent with our previous findings (*McQuade et al., 2020*; *Figure 7—figure supplement 1*). Addition of ADP to this system accelerated the scratch wound closure rates to the same extent in WT and TREM2 KO. In vivo, directed migration of microglia is often driven by gradients of ADP from dying or injured cells (*Haynes et al., 2006*; *Eyo et al., 2014*). Because no chemical gradient is formed in the scratch wound assay (*Liang et al., 2007*), we studied microglial

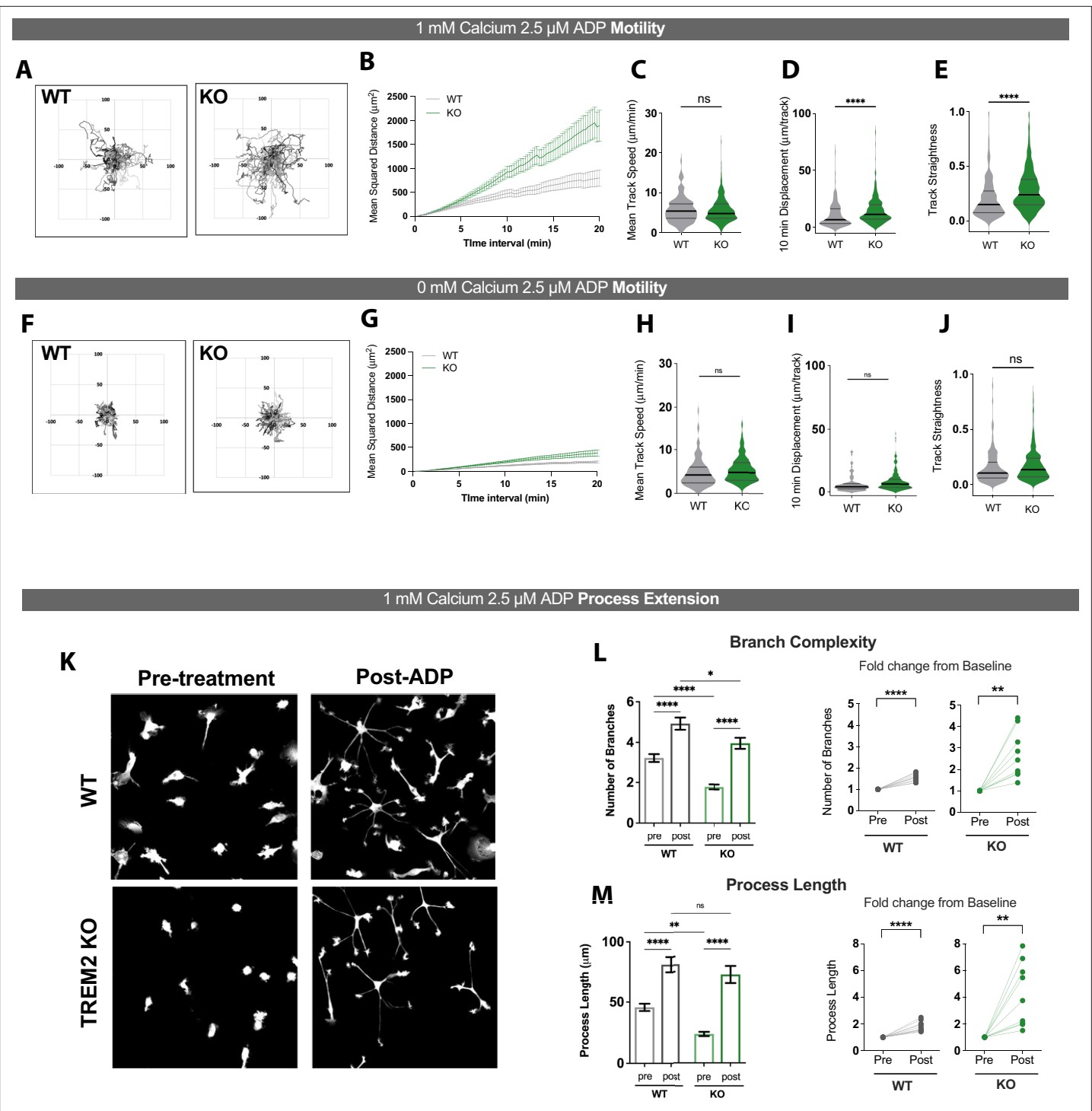

**Figure 5.** ADP-driven process extension and cell displacement are increased in TREM2 knockout (KO) induced pluripotent stem cell (iPSC)-microglia. (A–E) Motility of wild type (WT) (gray) and TREM2 KO (green) iPSC-microglia over 20 min following ADP addition in 1 mM $Ca^{2+}$-containing buffer. (A) Plots of track displacement in μm centered from point of origin at (0,0). (B) Mean squared displacement (MSD) vs. time. Mean cell track speeds (C), total track displacement in 10 min interval (D), and track straightness (E) for 130–327 cells, seven experiments, Student's *t*-test. (F–J) Same as (A–F) but in $Ca^{2+}$-free medium (125–279 cells, two experiments, Student's *t*-test). (K) Representative images of GFP-expressing WT (top) and TREM2 KO (bottom) iPSC-microglia, before and 30 min after 2.5 μM ADP addition. (L) Quantification of total number of branches per cell before and after ADP treatment (left) and paired dot plots showing fold change in branch number from pre-ADP levels (right). Each data point represents an imaging field in the paired plots. (M) Total process length before and after ADP treatment displayed as raw values per cell (left) and as fold change from baseline conditions per imaging field (right). For (L) and (M). n = 151–158 cells, WT; 133–167 cells, KO; 9–10 imaging fields, 3–4 experiments. One-way ANOVA

*Figure 5 continued on next page*

*Figure 5 continued*

with multiple comparisons for single-cell data, two-tailed paired *t*-test for the paired plots. Data shown as mean ± SEM (**B, G, L, M**) and as violin plots with mean, 25th and 75th percentile (**C– E, H–J**). p-Values indicated by ns, nonsignificant, *p<0.05, **p<0.01, and ****p<0.0001.

The online version of this article includes the following source data and figure supplement(s) for figure 5:

**Source data 1.** ADP-driven process extension and cell displacement are increased in TREM2 knockout (KO) induced pluripotent stem cell (iPSC)-microglia.

**Figure supplement 1.** Motility analysis in wild type (WT) and TREM2 knockout (KO) induced pluripotent stem cell (iPSC)-microglia.

**Figure supplement 2.** Comparison of process extension in wild type (WT) and TREM2 knockout (KO) microglia.

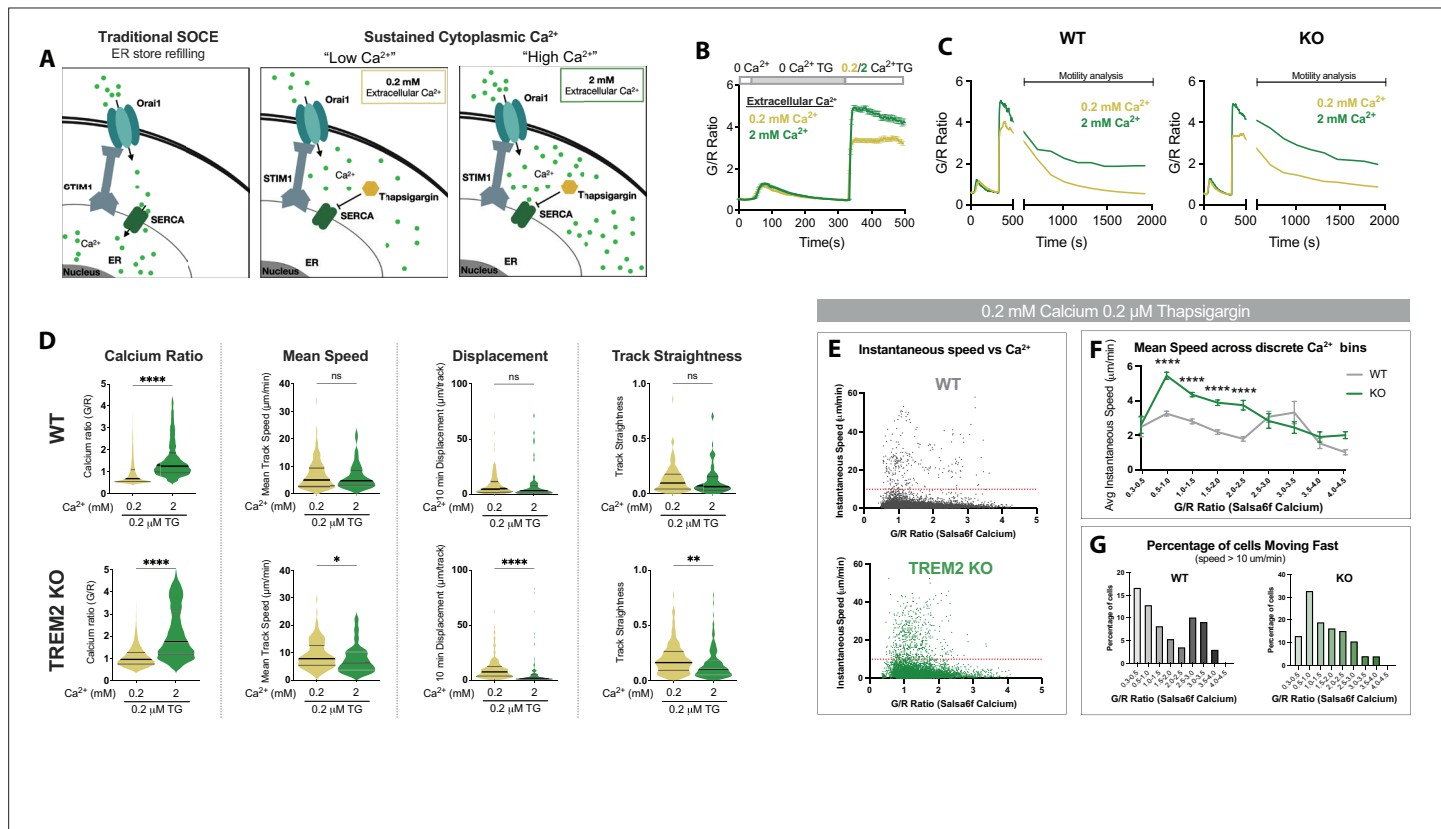

**Figure 6.** Cytosolic $Ca^{2+}$ levels tune microglial motility in TREM2 knockout (KO) cells. (**A**) Schematic of traditional store-operated $Ca^{2+}$ entry (SOCE) pathway with store refilling (left) and protocol for sustaining cytoplasmic $Ca^{2+}$ to 'low' and 'high' levels with 0.2 and 2 mM extracellular $Ca^{2+}$ and using thapsigargin (TG) to inhibit store refilling (right). (**B**) Average SOCE traces in wild type (WT) Salsa6f induced pluripotent stem cell (iPSC)-microglia showing changes in cytoplasmic $Ca^{2+}$ after addition of either 0.2 or 2 mM extracellular $Ca^{2+}$ (n = 78–110 cells). (**C**) Average change in cytoplasmic $Ca^{2+}$ levels in WT and TREM2 KO microglia over 25 min after SOCE activation. (**D**) Comparison of $Ca^{2+}$ levels and microglia motility in WT (top) and TREM2 KO (bottom) microglia. Cytosolic $Ca^{2+}$ levels indicated by instantaneous single-cell G/R ratio (n = 74–158 cells). Mean of instantaneous speeds, track displacement, and track straightness calculated as before in *Figures 3 and 4*. Yellow (0.2 mM Ca, TG), green (2 mM Ca, TG). Student's *t*-test ****p<0.0001; **p=0.0062; *p=0.432; ns > 0.9999. (**E**) Correlation of instantaneous $Ca^{2+}$ and instantaneous speed in WT and KO cells. Red line denotes 10 µm/s (cells above this threshold considered 'fast moving'). For WT: p<0.0001; *r* = –0.1316; number pairs = 5850. For KO: p<0.0001; *r* = –0.1433; number pairs = 6,063 (Spearman's correlation). (**F**) Mean speed of cells binned by instantaneous G/R $Ca^{2+}$ ratio (one-way ANOVA ****p<0.0001). Each data point is calculated for a bin increment of 0.5 G/R ratio. (**G**) Percentage of fast-moving cells quantified as a function of G/R $Ca^{2+}$ ratio. X-axis G/R ratios binned in increments of 0.5 as in (**F**). In (**E–G**), n = 78–100 cells. Data shown as mean ± SEM (**B, F**) and as violin plots with mean, 25th and 75th percentile (**D**). p-Values indicated by ns, nonsignificant, *p<0.05, **p<0.01, and ****p<0.0001.

The online version of this article includes the following source data and figure supplement(s) for figure 6:

**Source data 1.** Cytosolic $Ca^{2+}$ levels tune microglial motility in TREM2 knockout (KO) cells.

**Figure supplement 1.** Tracking cell motility and cytosolic $Ca^{2+}$ using Salsa6f-expressing induced pluripotent stem cell (iPSC) cell line.

**Figure supplement 2.** Motility analysis with varying $Ca^{2+}$.

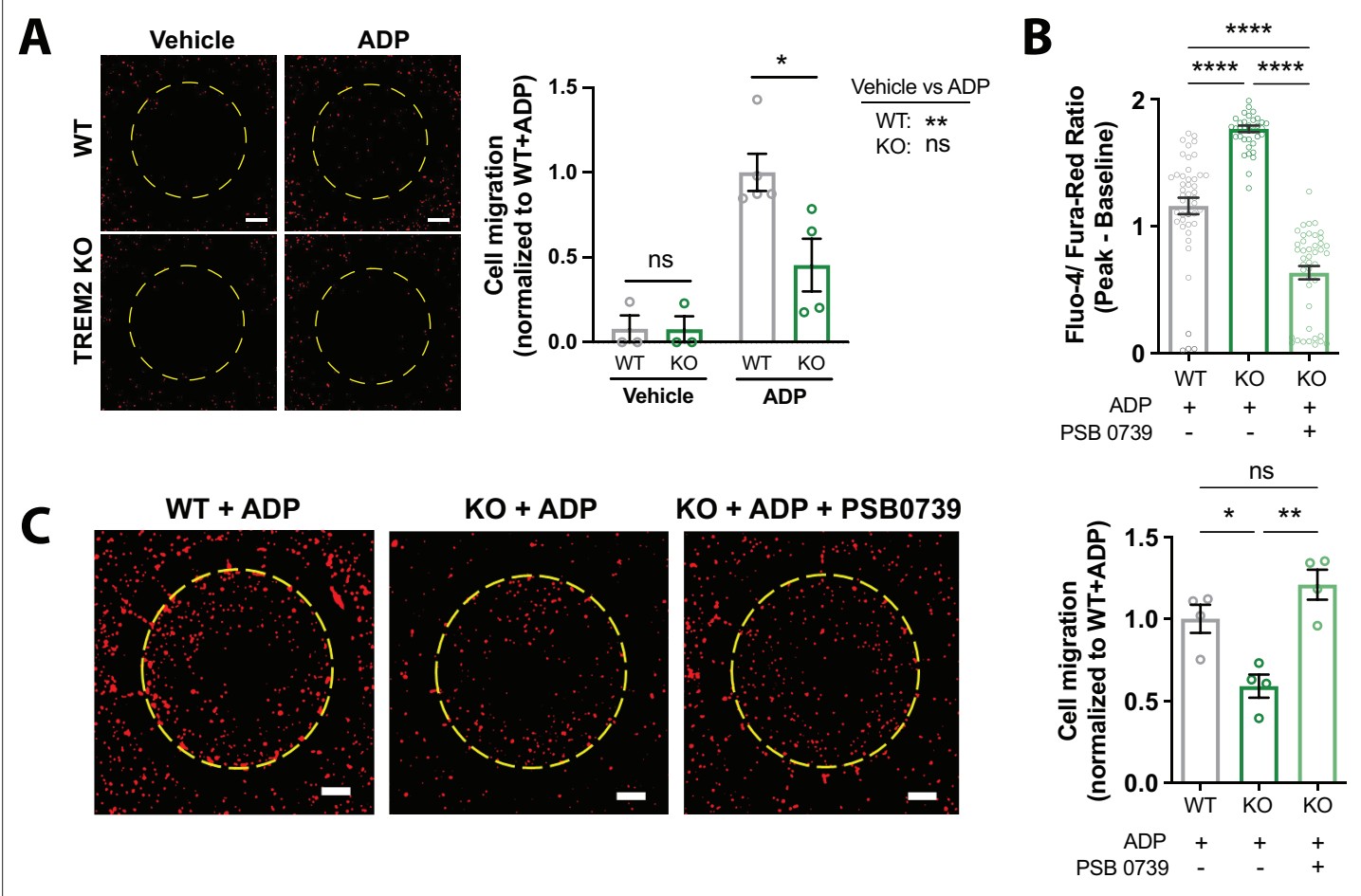

**Figure 7.** Migration deficits in TREM2 knockout (KO) microglia are rescued by inhibition of purinergic signaling. (**A**) Migration toward ADP in a two-chamber microfluidic device. Representative images of RFP-expressing microglia that migrated into the central chamber 3 days after 100 ng/mL ADP addition. Dotted circle delineates separation of inner and outer chambers. Scale bar = 500 µm. Quantification of microglial migration (right panel). Migrated cell counts are normalized to wild type (WT) cells treated with ADP (n = 3–4 experiments; one-way ANOVA with multiple comparisons). (**B**) Baseline-subtracted peak ratiometric $Ca^{2+}$ signal in response to 2.5 µM ADP in 1 mM extracellular $Ca^{2+}$, and in the presence or absence of 10 µM PSB 0739 (44 cells, WT; 39–43 cells, KO; representative of three independent experiments; one-way ANOVA with multiple comparisons). (**C**) Two-chamber migration to 100 ng/mL ADP with or without 10 µM PSB 0739. Values are normalized to WT cells with ADP (n = 3–4 experiments; one-way ANOVA with multiple comparisons). Representative images shown on the left. Scale bar = 500 µm. Data shown as mean ± SEM. p-Values indicated by ns, nonsignificant, *p<0.05, **p<0.01, and ****p<0.0001.

The online version of this article includes the following source data and figure supplement(s) for figure 7:

**Source data 1.** Migration deficits in TREM2 knockout (KO) microglia are rescued by inhibition of purinergic signaling.

**Figure supplement 1.** TREM2 wild type (WT) and knockout (KO) close scratch wound at similar rates.

chemotaxis toward ADP over a stable gradient using two-chamber microfluidic devices. Consistent with previous findings, WT iPSC-microglia directionally migrated up the concentration gradient of ADP, resulting in higher numbers of cells within the central chamber (*McQuade et al., 2020*; *Park et al., 2018*). In the absence of a chemotactic cue, this directional migration was lost (*Figure 7A*). This assay revealed a deficit of chemotaxis in TREM2 KO microglia (*Figure 7A*), mirroring reports that TREM2 KO microglia are unable to migrate toward amyloid plaques in AD (*Cheng-Hathaway et al., 2018*; *McQuade et al., 2020*; *Meilandt et al., 2020*). Given that ADP hypersensitivity in TREM2 KO cells is driven by increased expression of P2Y receptors, we examined the effects of dampening P2Y signaling to WT levels. Treatment with the $P2Y_{12}$ receptor antagonist, PSB 0739, reduced $Ca^{2+}$ responses in TREM2 KO cells and rescued the migration deficit in the chemotaxis assay (*Figure 7B and C*). These results link the increased $Ca^{2+}$ signals and altered motility characteristics evoked by ADP

in TREM2 KO cells to microglial chemotaxis toward areas of tissue damage, a vital functional response in microglia.

## Discussion

This study focuses on two aims: understanding the roles of purinergic signaling in regulating human microglial motility behavior and elucidating the impact of TREM2 loss of function on this $Ca^{2+}$ signaling pathway. We find that sustained $Ca^{2+}$ influx in response to ADP regulates microglial process extension, motility speed, and turning behavior. A key observation in our study is that microglia lacking TREM2 are highly sensitive to ADP-mediated signaling and show exaggerated cytoplasmic $Ca^{2+}$ responses. Using novel iPSC-microglia lines that express a ratiometric, genetically encoded $Ca^{2+}$ probe, Salsa6f, we found that the motility characteristics of human WT and TREM2 KO microglia are differentially tuned by $Ca^{2+}$ signaling. Informed by these discoveries, we were able to rescue chemotactic deficiencies in TREM2 KO microglia by dampening purinergic receptor signaling.

We provide several lines of evidence to show that hyper-responsiveness to purinergic ADP signaling in TREM2 KO microglia is driven primarily by increased purinergic $P2Y_{12}$ and $P2Y_{13}$ receptor expression: (1) $Ca^{2+}$ response is completely abrogated in the presence of $P2Y_{12}$ and $P2Y_{13}$ receptor inhibitors; (2) RNA-sequencing data shows significant increase in expression of $P2Y_{12}$ and $P2Y_{13}$ receptor transcripts but minimal fold change in other regulators of $Ca^{2+}$ signaling (IP3R, STIM, Orai, SERCA, and PMCA); and (3) labeling of surface $P2Y_{12}$ receptors shows greater PM expression in the TREM2 KOs. Furthermore, functional assays rule out any role for $Ca^{2+}$ clearance mechanisms or any difference in maximal $IP_3$ and SOCE activity as a cause of increased sustained $Ca^{2+}$ signal in TREM2 KO cells. Mechanistically, this increase in $Ca^{2+}$ signals is driven by enhanced $IP_3$-mediated ER store release coupled to SOCE. Indeed, based on the dose–response curves for peak ADP-$Ca^{2+}$ responses in $Ca^{2+}$-free buffer, TREM2 KO cells have an $EC_{50}$ at least 10-fold lower than WT cells. As a functional consequence, TREM2 KO microglia exhibit a defect in turning behavior and show greater displacement over time despite moving with similar speeds as the WT cells. The increased frequency in turning in WT microglia (relative to TREM KO cells) reflects greater canceling of the velocity vectors, which take the direction of motility into account. This restricts cell motility to more confined regions, potentially allowing for more frequent path correction. It is important to note that these motility differences with ADP are observed after acute treatment and in the absence of any gradient.

Interestingly, deletion of TREM2 had no significant impact on scratch wound closure rates, over a time scale of 24 hr in the presence of a constant concentration of ADP (*Ilina and Friedl, 2009*). However, we find in a directional chemotaxis assay toward a gradient of ADP concentration that TREM2 KO cells are unable to migrate as efficiently as WT cells, concordant with previous studies showing reduced migration of TREM2 KO cells toward Aβ plaques (*McQuade et al., 2020*). Enhanced ADP signaling likely abolishes the ability of TREM2 KO cells to distinguish gradations of the agonist, and this loss of gradient sensing results in an inability to perform directed migration. We speculate that increased ADP $Ca^{2+}$ signaling in TREM2 KO cells may result in $Ca^{2+}$ signaling domains that are no longer restricted to the cell region near to the highest ADP concentrations and disrupt the polarity of key signaling molecules that drive directed cell motility.

The amplitude and duration of $Ca^{2+}$ signals shape specificity of downstream cellular responses. Our experiments with ADP in $Ca^{2+}$-free medium revealed that a transient $Ca^{2+}$ signal is insufficient to induce microglial motility in either WT or TREM2 KO cells. Previous studies have shown that mouse microglia with genetic deletion of STIM1 or Orai1 also show defects in cell migration to ATP (*Michaelis et al., 2015*; *Lim et al., 2017*), likely because diminished SOCE renders them unable to sustain $Ca^{2+}$ signals in response to ATP. The dependence of motility on prolonged purinergic $Ca^{2+}$ signals may thus be a general feature of microglia. In contrast, a $Ca^{2+}$ transient can initiate some process extension in TREM2 KO but not in WT microglia, suggesting a threshold for ADP signaling that is reached in KO but not WT cells, and highlighting subtle differences in the $Ca^{2+}$ requirement for motility and process extension in TREM2 KO microglia.

To directly monitor $Ca^{2+}$ signaling and motility simultaneously in individual cells, we developed a novel iPSC-microglia cell line expressing a genetically encoded, ratiometric $Ca^{2+}$ indicator Salsa6f, a GCaMP6f-tdTomato fusion protein. Because Salsa6f allows simultaneous measurement of $Ca^{2+}$ signal and tracking of processes, this Salsa6f iPSC line is likely to be a useful tool to dissect the relationship between $Ca^{2+}$ signaling and the function of various iPSC-derived human cell types, including neurons,

astrocytes, and microglia. In addition, this line may be readily xenotransplanted for use with human/ microglia chimeric models to examine functional Ca$^{2+}$ responses to injury and pathology in vivo. Using Salsa6f-expressing microglia, we uncovered critical differences in how Ca$^{2+}$ levels tune motility in WT and TREM2 KO microglia. By tracking instantaneous velocity at the same time as Salsa6f Ca$^{2+}$ ratios in individual cells, we found that TREM2 KO cell motility showed a greater sensitivity to changes in cytosolic Ca$^{2+}$ levels with significantly higher speeds than WT cells at lower Ca$^{2+}$ and a more dramatic reduction in cell speed at high Ca$^{2+}$ levels. It is possible that high cytosolic Ca$^{2+}$ serves as a temporary STOP signal in microglia similar to its effects on T cells (*Negulescu et al., 1996*); we further speculate that TREM2 KO cells may be more subject to this effect with ADP, given the higher expression of P2RY$_{12}$ and P2Y$_{13}$ receptors. Accordingly, reducing cytosolic Ca$^{2+}$, resulted in increased mean speed, displacement, and straighter paths for TREM2 KO iPSC-microglia, but had no effect on these motility metrics in WT cells, suggesting that TREM2 KO cells may display a greater dynamic range in regulating their motility in response to sustained Ca$^{2+}$ elevations. Consistent with this observation, chemotaxis in TREM2 KO cells was restored by partially inhibiting P2Y$_{12}$ receptors. In response to neurodegenerative disease, microglia downregulate P2Y$_{12}$ receptors (*Krasemann et al., 2017*; *Sala Frigerio et al., 2019*; *Lou et al., 2016*). Active regulation of purinergic receptor expression is critical for sensing ADP gradients and decreasing motility near the chemotactic source. In vivo studies (*Hasselmann et al., 2019*; *Krasemann et al., 2017*; *McQuade et al., 2020*) suggest that TREM2 KO microglia are unable to downregulate P2Y receptor expression upon activation, which may lead to the known chemotactic deficits in these cells.

The studies presented here provide evidence that reducing purinergic receptor activity may be clinically applicable in Alzheimer's patients with TREM2 loss-of-function mutations (*Cheng-Hathaway et al., 2018*; *Piers et al., 2020*; *Parhizkar et al., 2019*). Pharmacologically targeting P2Y$_{12}$ receptors to dampen both the Ca$^{2+}$-dependent (PLC) and -independent (DAG) arms of the GPCR signaling pathway may be useful to control microglial activation and motility. However, our results suggest that altering downstream Ca$^{2+}$ flux may be sufficient, and thus, CRAC (Orai1) channel blockers that would specifically inhibit the sustained Ca$^{2+}$ signals without affecting the initial Ca$^{2+}$ transient or the activation of DAG may provide a more targeted approach.

Currently, TREM2 activating antibodies are being examined in early stage clinical trials for AD (*Alector Inc, 2021*; *Wang et al., 2020*), making it critically important to understand the broad consequences of TREM2 signaling. Therefore, an understanding of how TREM2 influences responses to purinergic signals and regulates cytosolic Ca$^{2+}$ in human iPSC-microglia is critical. Beyond TREM2, we have found that protective variants in MS4A6A and PLCG2 gene expression also decrease P2Y$_{12}$ and P2Y$_{13}$ receptor expression (unpublished data), suggesting that this mechanism of microglial activation could be common across several microglial AD risk loci.

In summary, deletion of TREM2 renders iPSC-microglia highly sensitive to ADP, leading to prolonged Ca$^{2+}$ influx, which increases cell displacement by decreasing cell turning. Despite this, TREM2 KO microglia show a defect in chemotaxis that is likely due to their inability to sense ADP gradients and make appropriate course corrections. Decreasing purinergic signaling in TREM2 KO microglia rescues directional chemotactic migration. We suggest that purinergic modulation or direct modulation of Ca$^{2+}$ signaling could provide novel therapeutic strategies in many AD patient populations, not solely those with reduced TREM2 function.

# Materials and methods

### Key resources table

| Reagent type (species) or resource | Designation | Source or reference | Identifiers | Additional information |
|---|---|---|---|---|
| Cell line (human) | WT iPSC-microglia | UCI ADRC iPSC Core | ADRC5; orgin: Blurton-Jones lab | iPSC-derived microglial line |
| Cell line (human) | TREM2 KO iPSC microglia | Blurton-Jones lab | ADRC5 Clone 28-18; orgin: Blurton-Jones lab | CRISPR-mediated knockout of TREM2 on the WT iPSC line |
| Cell line (human) | WT GFP-expressing iPSC-microglia | Coriell | AICS-0036; RRID:**CVCL_JM19** | iPSC-line with GFP tagged to αtubulin Originally developed by Dr. Bruce Conklin |

*Continued on next page*

*Continued*

| Reagent type (species) or resource | Designation | Source or reference | Identifiers | Additional information |
|---|---|---|---|---|
| Cell line (human) | TREM2 KO GFP-expressing iPSC-microglia | Blurton-Jones lab | GFP Clone 1 from above RRID | CRISPR-mediated knockout of TREM2 on the WT GFP⁺ iPSC line |
| Cell line (human) | WT RFP-expressing iPSC-microglia | Coriell | AICS-0031-035; RRID:**CVCL_LK44** | iPSC-line with RFP tagged to αtubulin Originally developed by Dr. Bruce Conklin |
| Cell line (human) | TREM2 KO RFP-expressing iPSC-microglia | Blurton-Jones lab | RFP Clone 6 from above RRID | CRISPR-mediated knockout of TREM2 on the WT RFP⁺ iPSC line |
| Cell line (human) | WT Salsa6f-expressing iPSC-microglia | UCI ADRC iPSC Core | ADRC76 Clone 8; orgin: Blurton-Jones lab | iPSC-line expressing a GCaMP6f-tdTomato fusion construct (Salsa6f) |
| Cell line (human) | TREM2 KO Salsa6f-expressing iPSC microglia | Blurton-Jones lab | ADRC76 Clone 8, Clone 98; orgin: Blurton-Jones lab | CRISPR-mediated knockout of TREM2 on the WT Salsa6f⁺ iPSC line |
| Cell line (human) | Orai1 KO iPSC microglia | Blurton-Jones lab | ADRC76; orgin: Blurton-Jones lab | CRISPR-mediated knockout of Orai1 on the WT ADRC76 iPSC line |
| Transfected construct (transgene) | Salsa6f | Addgene | Plasmid# 140188; RRID:**Addgene_140188** | A genetically encoded calcium indicator with tdTomato linked to GCaMP6f by a V5 epitope tag. |
| Other | DMEM/F12, HEPES, no Phenol red | Thermo Fisher Scientific | 11038021 | Microglia differentiation cell culture medium |
| Other | TeSR-E8 | STEMCELL Technologies | 05990 | Stem cell culture medium |
| Other | StemDiff Hematopoietic kit | STEMCELL Technologies | 05310 | |
| Peptide, recombinant protein | Nonessential amino acids | Gibco | 11140035 | |
| Peptide, recombinant protein | GlutaMAX | Gibco | 35050061 | |
| Peptide, recombinant protein (human) | Insulin | Sigma | I2643 | |
| Peptide, recombinant protein | B27 | Gibco | 17504044 | |
| Peptide, recombinant protein | N2 | Gibco | A1370701 | |
| Peptide, recombinant protein | Insulin-transferrin-selenite | Gibco | 41400045 | |
| Peptide, recombinant protein | IL-34 | PeproTech | 200-34 | |
| Peptide, recombinant protein | TGFβ1 | PeproTech | 100-21 | |
| Peptide, recombinant protein | M-CSF | PeproTech | 300-25 | |
| Peptide, recombinant protein | CX3CL1 | PeproTech | 300-31 | |
| Peptide, recombinant protein | CD200 | Novoprotein | C311 | |
| Peptide, recombinant protein | Fibronectin | STEMCELL Technologies | 07159 | |
| Other | Matrigel | Corning | 356231 | |
| Other | ReLeSR | STEMCELL Technologies | 5872 | Human pluripotent stem cell selection and passing reagent |
| Other | Goat serum | Thermo Fisher Scientific | 10,000C | |

*Continued on next page*

*Continued*

| Reagent type (species) or resource | Designation | Source or reference | Identifiers | Additional information |
|---|---|---|---|---|
| Other | Fluorescent beta-amyloid 1–42 (647) | AnaSpec | AS64161 | |
| Other | pHrodo tagged zymosan A beads | Thermo Fisher Scientific | P35364 | |
| Other | pHrodo tagged *S. aureus* | Thermo Fisher Scientific | A10010 | |
| Other | Human Stem Cell Nucleofector kit 2 | Lonza | VPH-5022 | |
| Other | Alt-R CRISPR-Cas9 tracrRNA | IDTDNA | 107253 | |
| Other | Alt-R HiFi Cas9 Nuclease | IDTDNA | 1081061 | |
| Antibody | Anti-human IBA1 (rabbit monoclonal) | Wako | 019-19741; RRID:AB_839504 | (1:200) |
| Antibody | Goat anti-rabbit 555 (secondary antibody) | Thermo Fisher Scientific | A21429; RRID:AB_2535850 | (1:400) |
| Other | Human TruStain FcX | BioLegend | Cat# 422301 | Fc blocking solution 5 µL per test |
| Antibody | Brilliant Violet 421 anti-human P2RY12 Primary antibody (mouse monoclonal) | BioLegend | 392105; clone 16001E; RRID:AB_2783290 | (5 µL) per test |
| Antibody | Brilliant Violet 421 mouse IgG2a κ Isotype control mouse | BioLegend | 407117; clone MOPC-173; RRID:AB_2687343 | (5 µL) per test |
| Chemical compound, drug | Fluo-4 AM | Thermo Fisher Scientific | F14201 | |
| Chemical compound, drug | Fura-red AM | Thermo Fisher Scientific | F3021 | |
| Chemical compound, drug | Pluronic F-127 | Thermo Fisher Scientific | P3000MP | |
| Chemical compound, drug | Cal-520 AM | AAT Bioquest | 21130 | |
| Chemical compound, drug | Cal-590 AM | AAT Bioquest | 20510 | |
| Chemical compound, drug | ci-IP3/PM | SiChem | 6210 | Caged-inositol triphosphate analog |
| Chemical compound, drug | Hoeschst | Thermo Fisher Scientific | R37165 | |
| Chemical compound, drug | ADP | Sigma-Aldrich | A2754 | |
| Chemical compound, drug | ATP | Sigma-Aldrich | A9187 | |
| Chemical compound, drug | UTP | Sigma-Aldrich | U1006 | |
| Chemical compound, drug | PSB 0739 | Tocris | 3983 | |
| Chemical compound, drug | MRS 2211 | Tocris | 2402 | |
| Chemical compound, drug | Synta66 | Sigma-Aldrich | SML1949 | Orai channel inhibitor |
| Chemical compound, drug | 2-APB | Sigma-Aldrich | D9754 | |
| Chemical compound, drug | Gadolinium | Sigma-Aldrich | G7532 | |
| Chemical compound, drug | EGTA | Sigma-Aldrich | E8145 | |
| Chemical compound, drug | 1-Thioglycerol | Sigma-Aldrich | M6145 | |
| Chemical compound, drug | CloneR | STEMCELL Technologies | 05888 | Defined supplement for single-cell cloning of human iPS cells |
| Chemical compound, drug | Thiazovivin | STEMCELL Technologies | 72252 | ROCK inhibitor |

*Continued on next page*

*Continued*

| Reagent type (species) or resource | Designation | Source or reference | Identifiers | Additional information |
|---|---|---|---|---|
| Other | 35 mm glass-bottom dish | MatTek | P35G-1.5-14C | 1.5 coverslip, 14 mm glass diameter |
| Other | Incubation perfusion Lid for 35 mm dishes | Tokai Hit | LV200-D35FME | Perfusion lid with inlet and outlet |
| Other | Laser Scanning Confocal Microscope | Olympus | FV3000 | Equipped with Resonant Scanner, IX3-ZDC2 Z-drift compensator, 40× silicone oil objective, 20× air objective |
| Other | Stage Top Incubation System | Tokai Hit | STXG | Temperature and humidity control for FV3000 microscope stage |
| Other | Nikon Eclipse T*i* microscope system | Nikon | | Equipped with a 40× oil immersion objective (NA 1.3; Nikon) and an Orca Flash 4.0LT CMOS camera (Hamamatsu) |
| Other | Chemotaxis Assay Chamber | Hansang Cho Lab | | |
| Other | IncuCyte S3 Live-Cell Analysis System | Sartorius | | |
| Other | Essen Incucyte WoundMaker | Sartorius | 4493 | |
| Software, algorithm | GraphPad Prism 9.1.0 | | | Data analysis, statistical analysis |
| Software, algorithm | Fiji (ImageJ) | | | Image analysis |
| Software, algorithm | Incucyte 2020C | | | Image acquisition and analysis |
| Software, algorithm | Imaris 9.7.0 | | | Cell tracking and image analysis |
| Software, algorithm | Flika | | | Image analysis |
| Software, algorithm | DiPer Excel Macros | | PMID:25033209 | Data analysis, directional persistence |

## Generation of iPSCs from human fibroblasts

Human iPSC lines were generated by the University of California, Irvine Alzheimer's Disease Research Center (UCI ADRC) Induced Pluripotent Stem Cell Core from subject fibroblasts under approved Institutional Review Boards (IRB) and human Stem Cell Research Oversight (hSCRO) committee protocols. Informed consent was received from all participants who donated fibroblasts. Reprogramming was performed with nonintegrating Sendai virus in order to avoid integration effects. To validate the karyotype and identity of iPSC lines, cells were examined via Microarray-based Comparative Genomic Hybridization (aCGH, Cell Line Genetics). Sterility and confirmation of mycoplasma negativity was examined every 10 passages and proceeding experimentation via MycoAlert (Lonza). Pluripotency was verified by Pluritest Array Analysis and trilineage in vitro differentiation. Additional GFP- and RFP-αtubulin-expressing iPSC lines (AICS-0036 and AICS-0031-035) were purchased from Coriell and originally generated by Dr. Bruce Conklin. Each Coriell line is provided with a corresponding certificate of analysis that verifies the correct reporter sequence insertion site, lack of plasmid integration, growth rate, expression of pluripotency markers, normal karyotype, sterility including mycoplasma negative, and identity of line via short tandem repeat (STR). See here and here.

## CRISPR-mediated knockout of TREM2 and ORAI1

Genome editing to delete TREM2 was performed as in *McQuade and Blurton-Jones, 2021*. Briefly, iPSCs were nucleofected with ribonucleoprotein complex targeting the second exon of TREM2 and allowed to recover overnight. Transfected cells were dissociated with pre-warmed Accutase then mechanically plated to 96-well plates for clonal expansion. Genomic DNA from each colony was amplified and sequenced at the cut site. The amplification from promising clones was transformed via TOPO cloning for allelic sequencing. Knockout of TREM2 was validated by Western blotting (AF1828, R&D) and HTRF (Cisbio) (*McQuade et al., 2020*). A similar strategy was used to delete ORAI1 using an RNP complex of Cas9 protein coupled with a guide RNA (5′ cgctgaccacgactacccac) targeting the second exon of ORAI1. The resulting ORAI1 clones were then validated to exhibit a normal for karyotype, identity, pluripotency, and sterility via Microarray-based Comparative Genomic Hybridization (aCGH, Cell Line Genetics), tri-lineage differentiation, and MycoAlert mycoplasma testing.

## iPSC-microglia differentiation

iPSC-microglia were generated as described in *McQuade et al., 2018* and *McQuade and Blurton-Jones, 2021*. Briefly, iPSCs were directed down a hematopoietic lineage using the STEMdiff Hematopoietic kit (STEMCELL Technologies). After 10–12 days in culture, CD43+ hematopoietic progenitor cells were transferred into a microglia differentiation medium containing DMEM/F12, 2× insulin-transferrin-selenite, 2× B27, 0.5× N2, 1× GlutaMAX, 1× nonessential amino acids, 400 µM monothioglycerol, and 5 µg/mL human insulin. Media was added to cultures every other day and supplemented with 100 ng/mL IL-34, 50 ng/mL TGF-β1, and 25 ng/mL M-CSF (PeproTech) for 28 days. In the final 3 days of differentiation, 100 ng/mL CD200 (Novoprotein) and 100 ng/mL CX3CL1 (PeproTech) were added to culture.

## Confocal laser scanning microscopy

Unless otherwise stated, cells were imaged on an Olympus FV3000 confocal laser scanning inverted microscope equipped with high-speed resonance scanner, IX3-ZDC2 Z-drift compensator, 40× silicone oil objective (NA 1.25), and a Tokai-HIT stage top incubation chamber (STXG) to maintain cells at 37°C. To visualize Salsa6f, 488 nm and 561 nm diode lasers were used for sequential excitation of GCaMP6f (0.3% laser power, 450 V channel voltage, 494–544 nm detector width) and TdTomato (0.05% laser power, 450 V channel voltage, 580–680 nm detector width), respectively. Fluo-4 and Fura-red were both excited using a 488 nm diode laser (0.07% laser power, 500 V channel voltage, 494–544 nm detector width for Fluo-4; 0.07% laser power, 550 V channel voltage, 580–680 nm detector for Fura-red). Two high-sensitivity cooled GaAsP PMTs were used for detection in the green and red channels, respectively. GFP was excited using the same settings as GCaMP6f. Other image acquisition parameters unique to $Ca^{2+}$ imaging, microglia process, and cell motility analysis are indicated in the respective sections.

## Measurement of intracellular $Ca^{2+}$

### Cell preparation

iPSC-microglia were plated on fibronectin-coated (5 µg/mL) glass-bottom 35 mm dishes (MatTek, P35G-1.5-14C) overnight at 60% confluence. Ratiometric $Ca^{2+}$ imaging was done using Fluo-4 AM and Fura-red AM dyes as described previously (*McQuade et al., 2020*). Briefly, cells were loaded in microglia differentiation medium with 3 µM Fluo-4 AM and 3 µM Fura-red AM (Molecular Probes) in the presence of Pluronic Acid F-127 (Molecular Probes) for 30 min at room temperature (RT). Cells were washed with medium to remove excess dye, and 1 mM $Ca^{2+}$ Ringer's solution was added to the 35 mm dish before being mounted on the microscope for live-cell imaging. We note that iPSC-microglia are sensitive to shear forces and produce brief $Ca^{2+}$ signals in response to solution exchange that are dependent on extracellular $Ca^{2+}$, and that these are more prominent at 37°C. To minimize these confounding effects, cells were imaged at RT and perfusion was performed gently. Salsa6f-expressing iPSC-microglia were prepared for $Ca^{2+}$ imaging in the same way as conventional microglia, but without the dye loading steps. The following buffers were used for $Ca^{2+}$ imaging: (1) 1 or 2 mM $Ca^{2+}$ Ringer's solution comprising 155 mM NaCl, 4.5 mM KCl, 1 mM $CaCl_2$, 0.5 mM $MgCl_2$, 10 mM glucose, and 10 mM HEPES (pH adjusted to 7.4 with NaOH); (2) $Ca^{2+}$-free Ringer's solution containing 155 mM NaCl, 4.5 mM KCl, 1.5 mM $MgCl_2$, 10 mM glucose, 1 mM EGTA, 10 mM HEPES, pH 7.4. Live-cell imaging was performed as described earlier. Cells were treated with ADP as indicated in the 'Results' section.

### Data acquisition

Time-lapse images were acquired in a single Z-plane at 512 × 512 pixels (X = 318.2 µm and Y = 318.2 µm) and at 2–3 s time intervals using Olympus FV3000 software. Images were time averaged over three frames to generate a rolling average and saved as .OIR files.

### Data analysis

Time-lapse videos were exported to Fiji-(ImageJ; https://imagej.net/Fiji), converted to TIFF files (16-bit), and background-subtracted. Single-cell analysis was performed by drawing ROIs around individual cells in the field, and average pixel intensities in the green and red channels were calculated for each ROI at each time point. GCaMP6f/ TdTomato (G/R Ratio) and Fluo-4/Fura-red ratio was then

obtained to further generate traces showing single-cell and average changes in cytosolic $Ca^{2+}$ over time. Single-cell ratio values were used to calculate peak $Ca^{2+}$ signal and responses at specific time points after agonist application as previously reported (*Jairaman and Cahalan, 2021*). Peak $Ca^{2+}$ signal for each cell was baseline-subtracted, which was calculated as an average of 10 minimum ratio values before application of agonist. SOCE rate was calculated as $\Delta(ratio)/\Delta t(s^{-1})$ over a 10 s time frame of maximum initial rise after $Ca^{2+}$ addback. Area under the curve (AUC) was calculated using the AUC function in GraphPad Prism.

## Microglia process extension analysis

### Data acquisition

GFP-expressing iPSC-microglia were plated overnight on 35 mm glass-bottom dishes at 40–50% confluence. Cells were imaged by excitation of GFP on the confocal microscope at 37°C as described earlier. To study process extension in response to ADP, two sets of GFP images were obtained for each field of view across multiple dishes: before addition of ADP (baseline) and 30 min after application of ADP. Images were acquired as a Z-stack using the Galvo scanner at Nyquist sampling. Adjacent fields of view were combined using the Stitching function of the Olympus FV3000 Software and saved as .OIR files.

### Process analysis

The basic workflow for microglia process analysis was adapted from *Morrison et al., 2017*. Image stacks (.OIR files) were exported to Fiji (ImageJ) and converted into 16-bit TIFF files using the Olympus Viewer Plugin (https://imagej.net/OlympusImageJPlugin). Maximum intensity projection (MIP) image from each Z-stack was used for further processing and analysis. MIP images were converted to 8-bit grayscale images, to which a threshold was applied to obtain 8-bit binary images. The same threshold was used for all sets of images, both before and after ADP application. Noise reduction was performed on the binary images using the Process -> Noise -> Unspeckle function. Outlier pixels were eliminated using Process -> Noise -> Outliers function. The binary images were then skeletonized using the Skeletonize2D/3D Plugin for ImageJ (https://imagej.net/plugins/skeletonize3d). Sparingly, manual segmentation was used to separate a single skeleton that was part of two cells touching each other. The Analyze Skeleton Plugin (https://imagej.net/plugins/analyze-skeleton/) was then applied to the skeletonized images to obtain parameters related to process length and number of branches for each cell in the imaging field. Processes were considered to be skeletons > 8 µm. The data was summarized as average process length and number of branches, before and after ADP application for a specific imaging field, normalized to the number of cells in the field that allowed for pairwise comparison. Additionally, single-cell data across all experiments were also compared in some instances.

## IP$_3$ uncaging

Whole-field uncaging of i-IP$_3$, a poorly metabolized IP$_3$ analog, was performed as previously described (*Lock et al., 2016*) with minor modifications. Briefly, iPSC-microglia were loaded for 20 min at 37°C with either Cal520 AM or Cal590 AM (5 µM, AAT Bioquest), and the cell-permeable, caged i-IP$_3$ analog ci-IP$_3$/PM (1 µM, SiChem) plus 0.1% Pluronic F-127 in Microglia Basal Medium. Cells were washed and incubated in the dark for further 30 min in a HEPES-buffered salt solution (HBSS) whose composition was (in mM) 135 NaCl, 5.4 KCl, 1.0 MgCl2, 10 HEPES, 10 glucose, 2.0 CaCl$_2$, and pH 7.4. Intracellular $Ca^{2+}$ ($[Ca^{2+}]_i$) changes were imaged by employing a Nikon Eclipse T*i* microscope system (Nikon) equipped with a 40× oil immersion objective (NA 1.3; Nikon) and an Orca Flash 4.0LT CMOS camera (Hamamatsu). Cal520 or Cal590 were excited by a 488 or a 560 nm laser light source (Vortran Laser Technologies), respectively. i-IP3 uncaging was achieved by uniformly exposing the imaged cells to a single flash of ultraviolet (UV) light (350–400 nm) from a xenon arc lamp. UV flash duration, and thus the amount of released i-IP$_3$ was set by an electronically controlled shutter.

Image acquisition was performed by using Nikon NIS (Nikon) software. After conversion to stack TIFF files, image sequences were analyzed with Flika, a custom-written Python-based imaging analysis software (https://flika-org.github.io/; *Ellefsen et al., 2014*). After background subtraction, either Cal520 or Cal590 fluorescence changes of each cell were expressed as $\Delta F/F_0$, where $F_0$ is the basal fluorescence intensity and $\Delta F$ the relative fluorescence change ($F_x – F_0$). Data are reported as superplots

(*Lord et al., 2020*) of at least three independent replicates. Experiments were reproduced with two independent lines. Comparisons were performed by unpaired nonparametric *t*-test.

## Immunocytochemistry

Cells were fixed with 4% paraformaldehyde for 7 min and washed 3× with 1× PBS. Blocking was performed at RT for 1 hr in 5% goat serum, 0.1% Triton5 X-100. Primary antibodies were added at 1:200 overnight 4°C (IBA1, 019-19741, FUJIFILM Wako). Plates were washed 3× before addition of secondary antibodies (goat anti-rabbit 555, Thermo Fisher Scientific) and Hoechst (Thermo Fisher Scientific). Images were captured on an Olympus FV3000RS confocal microscope with identical laser and detection settings. Images were analyzed with Imaris 9.7.0 software. We note that our attempt to verify Orai1 expression at the protein level was unsuccessful as the antibody used (Alomone, Cat# ALM-025, clone# 3F11/D10/B9) did not stain WT microglia in either immunostaining or western blot experiments.

### Flow cytometry

iPSC-derived microglia were seeded on fibronectin-coated 12-well plates at 200,000 cells/well. Cells were harvested and centrifuged in FACS tubes at $300 \times g$ for 5 min at 4°C. The cell pellet was subsequently resuspended in FACS buffer (1× PBS + 0.5% FBS). Fc receptors were blocked with a blocking buffer (BioLegend TruStain FcX in 1× PBS + 10% FCS). Cells were then incubated with Brilliant Violet 421-labeled anti-human $P2Y_{12}$ receptor antibody (clone S16001E, BioLegend, Cat# 392106) or with IgG2a isotype control antibody (clone MOPC-173, BioLegend, Cat# 400260) for 30 min at 4°C. Cells were washed, pelleted, and then resuspended in FACS buffer. Clone S16001E binds to the extracellular domain of the $P2Y_{12}$ and permits labeling of PM $P2Y_{12}$ receptors. Data were acquired using Novocyte Quanteon flow cytometer (Agilent) and analyzed using FlowJo analysis software (FlowJo v10.8.1 LLC Ashland, OR).

## Scratch wound assay

Nondirectional motility was analyzed using Essen Incucyte WoundMaker. iPSC-microglia were plated on fibronectin (STEMCELL Technologies) at 90% confluence. Scratches were repeated 4× to remove all cells from the wound area. Scratch wound confluency was imaged every hour until scratch wound was closed (15 hr). Confluence of cells within the original wound ROI was calculated using IncuCyte 2020C software.

## Imaris cell tracking

For motility assays, iPSC-microglia were tracked using a combination of manual and automatic tracking in Imaris 9.7.0 software. For videos of GFP lines, cells were tracked using spot identification. For videos of Salsa6f lines, surface tracking was used to determine ratiometric $Ca^{2+}$ fluorescence and motility per cell. In both conditions, tracks were defined by Brownian motion with the maximum distance jump of 4 µm and 10 frame disturbance with no gap filling. Tracks shorter than 3 min in length were eliminated from analysis. After automated track formation, tracks underwent manual quality control to eliminate extraneous tracks, merge falsely distinct tracks, and add missed tracks. After export, data was plotted in Prism 9.1.0 or analyzed in Excel using DiPer Macros for Plot_At_Origin (translation of each trajectory to the origin) and $MSD(t) = 4D(t-P(1-e^{(-t/P)}))$, where D is the diffusion coefficient, t is time, and P represents directional persistence time (time to cross from persistent directionality to random walk) (*Gorelik and Gautreau, 2014*). From Imaris, speed was calculated as instantaneous speed of the object (µm/s) as the scalar equivalent to object velocity. These values were transformed to µm / min as this time scale is more relevant for the changes we observed. Mean track speed represents the mean of all instantaneous speeds over the total time of tracking. 10 min displacement is calculated by (600) * (TDL/TD), where TDL is the track displacement length (distance between the first and last cell position) represented as TDL = p(n) - p(1) for all axes, where the vector p is the distance between the first and last object position along the selected axis, and TD is the track duration represented as TD = T(n) - T(1), where T is the time point of the first and final time point within the track. Frame-to-frame displacement is calculated as p(n) – p(n-1) for all the different frames in a cell track. Track straightness is defined as TDL/TL, where TDL is the track displacement as described above and TL is the track

length representing the total length of displacements within the track TL = sum from t = 2 to n of |p(t)-p(t-1)|.

## Generation of Salsa6f-expressing iPSC lines

iPSCs were collected following Accutase enzymatic digestion for 3 min at 37°C. 20,000 cells were resuspended in 100 µL nucleofection buffer from Human Stem Cell Nucleofector Kit 2 (Lonza). Salsa6f-AAVS1 SHL plasmid template (2 µg; Vector Builder) and RNP complex formed by incubating Alt-R S.p. HiFi Cas9 Nuclease V3 (50 µg; IDTDNA) was fused with crRNA:tracrRNA (IDTDNA) duplex for 15 min at 23°C. This complex was combined with the cellular suspension and nucleofected using the Amaxa Nucleofector program B-016. To recover, cells were plated in TeSR-E8 (STEMCELL Technologies) media with 0.25 µM thiazovivin (STEMCELL Technologies) and CloneR (STEMCELL Technologies) overnight. The following day, cells were mechanically replated to 96-well plates in TeSR-E8 media with 0.25 µM thiazovivin and CloneR supplement for clonal isolation and expansion. Plates were screened visually with a fluorescence microscope to identify TdTomato$^+$ clones. Genomic DNA was extracted from positive clones using Extracta DNA prep for PCR (Quantabio) and amplified using Taq PCR Master Mix (Thermo Fisher Scientific) to confirm diallelic integration of the Salsa6f cassette. A clone confirmed with diallelic Salsa6f integration in the AAVS1 SHL was then retargeted as previously described (*McQuade et al., 2020*) to knock out Trem2.

## Phagocytosis assay

Phagocytosis of transgenic iPSC-microglia was validated using IncuCyte S3 Live-Cell Analysis System (Sartorius) as in *McQuade et al., 2020*. Microglia were plated at 50% confluency 24 hr before substrates were added. Cells were treated with 50 µg/mL pHrodo tagged human AD synaptosomes (isolated as described in *McQuade et al., 2020*), 100 ng/mL pHrodo tagged zymosan A beads (Thermo Fisher Scientific), 100 ng/mL pHrodo tagged *Staphylococcus aureus* (Thermo Fisher Scientific), or 2 µg/mL fluorescent beta-amyloid (AnaSpec). Image masks for fluorescence area and phase were generated using IncuCyte 2020C software.

## Chemotaxis assay

iPSC-microglia were loaded into the angular chamber (2–5K cells/device) to test activation and chemotaxis toward the central chamber containing either ADP (100 ng/mL or 234 nM) or vehicle. When noted, PSB 0739 (10 µM) was added to both the central and angular chamber to inhibit P2Y$_{12}$ receptors. To characterize motility, we monitored the number of recruited microglia in the central chamber for 4 days under the fully automated Nikon TiE microscope (10× magnification; Micro Device Instruments, Avon, MA).

## Statistical analysis

GraphPad Prism (versions 6.01 and 8.2.0) was used to perform statistical tests and generate p-values. We used standard designation of p-values throughout the figures (ns, not significant or $p \geq 0.05$; $*p < 0.05$; $**p < 0.01$; $***p < 0.001$; $****p < 0.0001$). Traces depicting average changes in cytosolic Ca$^{2+}$ over time are shown as mean ± standard error of the mean (SEM). Accompanying bar graphs with bars depicting mean ± SEM provide a summary of relevant parameters (amplitude of Ca$^{2+}$ response, degree of store release, rate of Ca$^{2+}$ influx, etc.) as indicated. Details of the number of replicates and the specific statistical test used are provided in the individual figure legends.

## Acknowledgements

We thank Dr. Andy Yeromin for the development of Excel macros to analyze Imaris cell tracking. We would also like to thank Morgan Coburn for sharing Python scripts that aided in the organization of Imaris output files. This work was supported by T32 NS082174 and ARCS foundation (AM); the European Union's Horizon 2020 research and innovation program under the Marie Sklodowska-Curie grant agreement iMIND – no. 84166 (AG); NIH R01 NS14609 and AI121945 (MDC); NIH U01 AI160397 (SO); NRF 2020R1A2C2010285, 2020 M3C7A1023941, and NIH AG059236-01A1 (HC); NIH AG048099, AG056303, and AG055524 (MBJ); RF1DA048813 (MBJ and SG); UCI Sue & Bill Gross Stem Cell Research Center Seed Grant (SG); and a generous gift from the Susan Scott Foundation (MBJ). iPSC lines were generated by the UCI-ADRC iPS cell core funded by NIH AG066519. Experiments using the

GFP-expressing iPSC line AICS-0036 were made possible through the Allen Cell Collection, available from the Coriell Institute for Medical Research.

## Additional information

### Competing interests

Sunil Gandhi: is a co-founders of NovoGlia Inc. Mathew Blurton-Jones: is a co-inventor of patent application WO/2018/160496, related to the differentiation of pluripotent stem cells into microglia. Is a co-founders of NovoGlia Inc. The other authors declare that no competing interests exist.

### Funding

| Funder | Grant reference number | Author |
|---|---|---|
| National Institutes of Health | R01 NS14609 | Michael D Cahalan |
| National Institutes of Health | R01 AI121945 | Michael D Cahalan |
| National Institutes of Health | R01 AG048099 | Mathew Blurton-Jones |
| National Institutes of Health | R01 AG056303 | Mathew Blurton-Jones |
| National Institutes of Health | R01 AG055524 | Mathew Blurton-Jones |
| National Institutes of Health | core AG066519 | Mathew Blurton-Jones |
| National Institutes of Health | U01 AI160397 | Shivashankar Othy |
| National Institutes of Health | T32 NS082174 | Amanda McQuade |
| National Institutes of Health | RF1DA048813 | Sunil Gandhi |
| The Marie Sklodowska-Curie grant agreement iMIND | no. 84166 | Alberto Granzotto |
| National Research Foundation | 2020R1A2C2010285 | Hansang Cho |
| National Research Foundation | 2020 M3C7A1023941 | Hansang Cho |
| National Research Foundation | I21SS7606036 | Hansang Cho |
| National Institute of Health | AG059236-01A1 | Hansang Cho |
| UCI Sue & Bill Gross Stem Cell Research Center | Seed Grant | Sunil Gandhi |
| Susan Scott Foundation | gift | Mathew Blurton-Jones |

The funders had no role in study design, data collection and interpretation, or the decision to submit the work for publication.

### Author contributions

Amit Jairaman, Conceptualization, Formal analysis, Investigation, Methodology, Validation, Visualization, Writing – original draft, Writing – review and editing; Amanda McQuade, Conceptualization, Formal analysis, Funding acquisition, Investigation, Methodology, Validation, Visualization, Writing – original draft, Writing – review and editing; Alberto Granzotto, You Jung Kang, Formal

analysis, Investigation, Methodology, Writing – review and editing; Jean Paul Chadarevian, Investigation, Methodology; Sunil Gandhi, Funding acquisition, Resources, Supervision, Writing – review and editing; Ian Parker, Methodology, Resources, Writing – review and editing; Ian Smith, Formal analysis, Investigation, Writing – review and editing; Hansang Cho, Formal analysis, Funding acquisition, Investigation, Methodology, Resources, Supervision, Writing – review and editing; Stefano L Sensi, Funding acquisition, Methodology, Resources, Supervision; Shivashankar Othy, Conceptualization, Formal analysis, Funding acquisition, Investigation, Methodology, Project administration, Resources, Supervision, Writing – original draft, Writing – review and editing; Mathew Blurton-Jones, Conceptualization, Funding acquisition, Investigation, Methodology, Project administration, Resources, Supervision, Writing – original draft, Writing – review and editing; Michael D Cahalan, Conceptualization, Funding acquisition, Methodology, Project administration, Resources, Supervision, Writing – original draft, Writing – review and editing

## Author ORCIDs
Amit Jairaman (iD) http://orcid.org/0000-0002-5206-700X
Amanda McQuade (iD) http://orcid.org/0000-0001-5368-6788
Ian Smith (iD) http://orcid.org/0000-0001-9910-195X
Shivashankar Othy (iD) http://orcid.org/0000-0001-6832-5547
Mathew Blurton-Jones (iD) http://orcid.org/0000-0002-7770-7157
Michael D Cahalan (iD) http://orcid.org/0000-0002-4987-2526

## Ethics
Human subjects: Human iPSC lines were generated by the University of California Alzheimer's Disease Research Center (UCI ADRC) stem cell core. Subject fibroblasts were collected under approved Institutional Review Boards (IRB) and human Stem Cell Research Oversight (hSCRO) committee protocols. Informed consent was received for all participants.

## Decision letter and Author response
Decision letter https://doi.org/10.7554/eLife.73021.sa1
Author response https://doi.org/10.7554/eLife.73021.sa2

# Additional files

## Supplementary files
• Transparent reporting form

## Data availability
RNA sequencing data referenced in Figure 1- figure supplement 2 is available through Gene Expression Omnibus: GSE157652.

The following dataset was generated:

| Author(s) | Year | Dataset title | Dataset URL | Database and Identifier |
|---|---|---|---|---|
| McQuade A | 2020 | Transcriptomic and functional deficits in human TREM2-/- microglia impair response to Alzheimer's pathology in vivo [RNA-seq] | http://www.ncbi.nlm.nih.gov/geo/query/acc.cgi?acc=GSE157652 | NCBI Gene Expression Omnibus, GSE157652 |

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
