## [Editor Report]

Overall, this is a significant advance in the field of microglial regulation by calcium signaling.

---

## [Decision Letter]

**Decision letter after peer review:**

Thank you for submitting your article "TREM2 regulates purinergic receptor-mediated calcium signaling and motility in human iPSC-derived microglia" for consideration by *eLife*. Your article has been reviewed by 3 peer reviewers, one of whom is a member of our Board of Reviewing Editors, and the evaluation has been overseen by Richard Aldrich as the Senior Editor. The following individual involved in review of your submission has agreed to reveal their identity: Mohamed Trebak (Reviewer #2).

The reviewers have discussed their reviews with one another, and the Reviewing Editor has drafted this to help you prepare a revised submission. Reviewers had concerns on two broad aspects of the paper: the calcium signaling studies, and the results relating to cell motility. In addition, other concerns were raised which are described in the individual reviewer comments.

Essential revisions:

1) A more rigorous examination of protein expression should be performed with antibody validation to determine if there are compensatory changes at the protein level of other relevant Ca^2+^ signaling proteins (Figure 1 Suppl2) that could affect the results. Westerns should be performed for STIM1 and STIM2, Orai1 and IP3R 1/2/3, for which there are reliable Abs. Although the IF as performed for Orai1 is acceptable, the authors do not show positive and negative controls and a thorough validation of this Orai1Ab in IF. This is strongly recommended given the poor quality of Orai1 Abs in general. Likewise, staining for P2YR12 appears internal and the Ab should be validated. A WB on P2YR12 and P2RY13 in addition would help.

2) Important: It is clear that the TREM2 KO cells have significantly lower STIM1 expression, which would be expected to impact SOCE under conditions of physiological agonists stimulation (ADP, ATP). The conclusion based on maximal activation with thapsigargin does not address this issue. What is the impact of reduced STIM1 expression on the Ca signals and motility?

3) A Ca^2+^ off/on protocol with ADP (Figure 1) as done for thapsigargin should be performed. In Figure 2B, ADP was applied in the absence of external Ca^2+^, but sadly those recordings did not subsequently replenish Ca^2+^ to gauge Ca^2+^ entry. Low concentrations of Gd3+ (which should not alter Ca^2+^ release) reduce the initial Ca^2+^ release peak, suggesting that there is a contribution of Ca^2+^ entry to this initial response. Also, it would have been helpful to include in Figure 1 Suppl1 the control (non KO cells) in these experiments for reference.

4) The conclusion that SOCE is mediated by Orai1 should be validated with a genetic approach as 2-APB and La3+ are non-specific.

5) Conversely, it is not clear if store content is depleted to a greater extent following ADP receptor stimulation in the KO cells. Assess ER calcium content in response to ADP in WT and KO cells or with an ionomycin pulse.

6) Figure 3: The WT data in this figure don't include KO cells. How do the TREM2 KOs compare to WT in the scratch wound closure assay? The KO cells seem to be tested, but only in the absence of ADP (Figure 4 Sup 1A). Is there a difference in wound closure after ADP treatment? This point should be evaluated to assess the effects of altered ADP Ca signaling on non-directional cell movement in a physiologically relevant paradigm.

7) Terminology: The term "Ca clamp" in the way it has been used here could be misconstrued by readers. Cellular Ca is not clamped. It varies within each cell depending on the activity of the pumps and other clearance mechanisms in the cell. The motility measurements are simply done after extracellular Ca is restored to the cells after thapsigargin treatment, at time points where the cytosolic Ca is lower in 0.2 mM Ca compared to 2 mM Ca. This phrasing (Ca clamp) needs to be changed to eliminate confusion that cellular calcium is "clamped" analogous to voltage-clamp.

8) Figure 6D TREM2 KOs (but not WT) cells show differences in motility between 0.2 mM and 2 mM extracellular Ca even after thapsigargin treatment with no ADP exposure. This is strong indication that motility differences in TREM2 KO cells are due to more than just alterations in purinergic signaling. (Consistent with lack of additional ADP effects in Figure 6 Supplement 1). What is the role of these intrinsic changes in cell motility in response to cellular Ca elevations in the presence and absence of nucleotides?

9) Figure 6D: The data for WT cells should be directly compared to the KOs at each extracellular Ca concentration to dissect out differences in KO cells at each Ca condition.

10) The data in Figure 4K should be reanalyzed to compare differences in the absolute levels of branch complexity (# of branches and process length) in the two groups.

11) The apparent disconnect between the larger mean square distance travelled in the absence of nucleotides in the KO with no change in the mean speed needs to be clarified. Is this due to cancelling of speed vectors with opposite directions?

*Reviewer #1 (Recommendations for the authors):*

1) Figure 3: The WT data in this figure don't include KO cells. How do the KOs compare to the WT in the scratch wound closure assay? The KO cells seem to be tested, but only in the absence of ADP (Figure 4 Sup 1A). Is there a difference in wound closure after ADP treatment? This point should be evaluated to assess the effects of altered ADP Ca signaling on non-directional cell movement in a physiologically relevant paradigm.

2) Figure 4: This reviewer is not convinced that reduced turning accounts for the similar speed in the KO despite the larger mean square distance travelled. If turning is reduced and the mean square distance is doubled in the KO, then effectively, the average distance travelled over time (speed) should be greater in the KO than in the WT. The logic of this inference and how it explains the data (similar average speed but much greater root mean square displacement in the KO) needs to be better clarified. Since the mean square distance is so much larger, it suggests that the speed vectors with opposite directions are cancelling out in the KO cells to yield the same average velocity (which could theoretically even occur from increased, not decreased turning).

3) Figure 4K: It seems that the baseline branch complexity is low to start with in the KOs, hence the fold change from baseline seems larger. From the images shown, there appears to be no difference following ADP treatment between the two groups. This data should be reanalyzed to compare differences in the absolute levels of branch complexity (# of branches and process length) in the two groups.

4) The use of the term Ca clamp is misleading in the way it has been used here. Cellular Ca is, in fact, not clamped, but changes within each cell depending on the activity of pumps and other calcium clearance mechanisms. The motility measurements are simply done after extracellular Ca is restored to cells after thapsigargin treatment, at time points where the cytosolic Ca is lower in 0.2 mM Cacompared to 2 mM Ca. This phrasing (Ca clamp) needs to be changed to eliminate confusion that cellular Ca levels are "clamped" to defined values analogous to voltage clamp.

5) Figure 6D: It is noteworthy that TREM2 KOs, but not WT cells, show differences in motility between 0.2 mM and 2 mM extracellular Ca even after thapsigargin treatment with no ADP exposure. This is strong suggestion that motility differences in TREM2 KO cells are due to more than just alterations in purinergic signaling. In fact, this inference is consistent by the lack of additional effects of adding ADP in Figure 6 Supplement 1. These intrinsic changes in cell motility in response to cellular Ca elevations need further clarification.

6) Figure 6D: Related to this figure, the data for WT cells should be directly compared to the KOs at each extracellular Ca concentration to dissect out differences in KO cells at each Ca condition.

Other:

– Stim1 is expressed at different levels between the WT and TREM2 KO. It has a significant p-value (Figure 1 Sup2D). This decrease could affect multiple aspects of microglia including cell motility in the presence of physiologically relevant agonists. This needs to be addressed.

– A technical point related to Figure 4 is that figures 4C and 4G should be labelled "velocity" rather than speed, as it seems the only way in which the mean squared distance can be twice as large in the KO but not the velocity if the speed vectors with opposite directions are cancelling out in the KO.

– The dose of ADP used in Figure 7 is extremely high. 100ng/mL (which is 234uM by my calculation). By contrast, Figure 4 used 2.5uM. The 100 fold higher concentration in the chemotaxis assay needs to be justified. Further, it should be determined whether the ADP used diffuses into both chambers might be expected. This super high dose of ADP may have non-physiological effects that could explain the seemingly differing phenotypes in Figures 4 and 7.

– Figure 5 is a great set of experiments to show that the Salsa6f does not affect end points of the study, but may be better as a supplementary figure rather than a main figure. If the paper is about the TREM2 KO versus WT then the main figures should stick with looking at differences between the two with maybe a sub figure to show validation.

*Reviewer #2 (Recommendations for the authors):*

– A more rigorous approach could be used to document the presence or absence of compensatory changes at the protein protein level of other Ca^2+^ signaling proteins (Figure 1 Suppl2). Preferably Westerns should be performed for STIM1 and STIM2, Orai1 and IP3R 1/2/3, for which there are reliable Abs. Although IF as performed for Orai1 is acceptable, the authors do not show positive and negative controls and a thorough validation of this Orai1Ab in IF.

– Along the same lines as comment above, Staining for P2YR12 appears internal. Has the Ab been validated? Can authors perform a WB on P2YR12 and P2RY13 in addition?

– Did the authors check on whether the expression of SECRA is altered at the protein level? Did the authors perform CRAC channel recordings in Control and TREM2 KO cells?

– It is quite surprising that the authors did not opt to use the Ca^2+^ off/on protocol with ADP (Figure 1) as they have done with thapsigargin. In Figure 2B, ADP was applied in the absence of external Ca^2+^, but sadly those recordings did not subsequently replenish Ca^2+^ to gauge Ca^2+^ entry. One can rationalize the result of 2-APB (Figure 1 Suppl1), a non-specific channel blocker that also inhibits IP3Rs. However, the fact that low concentrations of Gd3+ (which should not alter Ca^2+^ release) reduce the initial Ca^2+^ release peak, suggests that there is a contribution of Ca^2+^ entry to this initial response. Also, it would have been helpful to include in Figure 1 Suppl1 the control (non KO cells) in these experiments for reference.

Additional thoughts to the attention of the authors:

– The authors have speculated on the fact that increased cytosolic Ca^2+^ might lead to a "spill" outside the restricted Ca^2+^ nanodomain, thus disrupting polarity. Do the authors have any evidence of altered location of leading edge vs trailing edge proteins or focal adhesion proteins in TREM2 KO cells?

– Have the authors performed RNAseq on WT control and TREM2 KO cells? If so, does pathway analysis shows changes in the Cell motility genes between these two groups? This might offer some interesting clues to pursue in future studies.

*Reviewer #3 (Recommendations for the authors):*

This study by Jairaman et al. describes how iPSC-derived microglia exhibit exaggerated cytosolic Ca^2+^ responses to ADP stimulation in TREM2KO cells, and that this leads to a defect in turning behaviour and hence no directed migration to a chemotactic signal.

Overall, the experiments are well conducted, carefully controlled and the findings are new and exciting. The authors nicely dissect out the underlying molecular basis for the exaggerated Ca^2+^ responses to ADP and then extend their findings to cell movement and directed migration. Given the substantial body of evidence linking microglia to the pathogenesis of Alzheimer's disease, and the role for TREM2, the work by Jairaman et al. is of translational significance. As an aside, the introduction of the calcium-sensitive reporter Salsa6F is a welcome new tool in the arsenal for recording cytosolic calcium. Overall, this is an elegant, novel and important study. Nevertheless, I have a few comments/suggestions.

1. The authors argue that the increased Ca^2+^ plateau to ADP in TREM2KO cells is due to enhanced Ca^2+^ release from the stores ie greater store depletion. Evidence is presented that maximal SOCE is not compromised in TREM2KO cells, that store content under resting conditions is unaffected, InsP3R activities are similar. But the authors do not demonstrate that store content falls more following ADP receptor stimulation in the KO cells. This could be shown by directly measuring ER calcium to ADP in WT and KO cells or by applying ADP in Ca-free solution and then assessing store content with an ionomycin pulse. The ionomycin response should be smaller in the TREM2KO cells, after ADP exposure.

2. The authors suggest that an increase in P2Y12 and P2Y13 receptor expression in the KO cells accounts for the increased Ca^2+^ release from the ER. However, the increase in P2Y12 protein levels is modest at best, and the increase for P2y13 is less than 2-fold. The authors' point would be strengthened by showing InsP3 levels are increased in the KO cells compared with WT ones, for the same dose of ADP. Single cell InsP3 probes are available. Alternatively, a population measurement could be carried out.

3. The evidence for an involvement of SOCE is based on two rather non-specific inhibitors (2-APB and Gd3+). P2YRs, like other GPCRs, activate TRPC proteins and these Ca^2+^-permeable channels could contribute to the Ca^2+^ plateau. The authors should therefore knock down Orai1 or at least use more selective inhibitors such as Synta66 or the GSK compound.

4. The authors should include some controls to show the P2Y antibodies are indeed specific. For example, they could use a cell line that lacks P2Y12/13. In Figure 2I, J, what happens to the ADP response when both P2Y inhibitors are present at the same time? It would be important to show that the combination of P2Y12 and P2Y13 suppress ADP responses fully.

5. The authors should add a justification for the concentrations of agonists they have used (ADP, ATP, UTP). A dose-response curve is included for ADP but a comment on the doses selected would be helpful.

6. It is nicely shown that Ca^2+^ influx in TREM2 KO cells leads to microglia motility and process extension to a greater extent than WT cells. This is attributed to the increased SOCE. If so, then one might expect raising external Ca^2+^ in WT cells to have the same effect. Is this the case?

7. Is anything known mechanistically how the exaggerated Ca^2+^ signal leads to a chemotaxis defect? The authors' data would suggest that chemotaxis might have a bell-shaped dependence on cytosolic Ca^2+^, with too little or too much impeding migration towards a cue.

8. Do the authors know whether the effects of SOCE are mediated through a local Ca^2+^ rise or via a global cytosolic Ca elevation?

9. Perhaps I missed something but Figure 1 S2(D) seems to show a significant decrease in STIM1 levels in the KO cells.

10. The authors present SOCE as the peak signal remaining after 5 minutes. It is not clear whether the value has been subtracted from the pre-stimulation levels. In some graphs, the signal after 5 minutes looks the same as the resting level, but the bar charts show higher values.

---

## [Author Response]

Reviewers had concerns on two broad aspects of the paper: the calcium signaling studies, and the results relating to cell motility. In addition, other concerns were raised which are described in the individual reviewer comments.Essential revisions:1) A more rigorous examination of protein expression should be performed with antibody validation to determine if there are compensatory changes at the protein level of other relevant Ca^2+^ signaling proteins (Figure 1 Suppl2) that could affect the results. Westerns should be performed for STIM1 and STIM2, Orai1 and IP3R 1/2/3, for which there are reliable Abs. Although the IF as performed for Orai1 is acceptable, the authors do not show positive and negative controls and a thorough validation of this Orai1Ab in IF. This is strongly recommended given the poor quality of Orai1 Abs in general. Likewise, staining for P2YR12 appears internal and the Ab should be validated. A WB on P2YR12 and P2RY13 in addition would help.

Our approach in this study relies on functional Ca^2+^ imaging readouts in combination with pharmacological and genetic tools to assess P2Y receptor sensitivity to agonists and antagonists and downstream signaling events that include IP_3_R activation, ER store-release, and SOCE in WT and TREM2 KO microglia. Given our focus on functional data that are concordant with RNA expression results and point toward the differential expression of P2Y receptors, we did not see a strong rationale for further dissecting the expression of various STIM, Orai, IP_3_R or pump isoforms at the protein level. New data in the paper strengthen this conclusion. These include the ionomycin and TG-pulse experiments (Figure 3E, and Figure 3—figure supplement 2A, B) together with the ADP-Ca^2+^ addback experiments (Figure 3D), showing that ADP depletes ER Ca^2+^ stores to a greater extent in TREM2 KO cells. The new data reinforce the conclusion that increased expression of P2Y receptors in TREM2 KO microglia drives downstream calcium signaling events.

We concur with the reviewer that appropriate antibody controls are needed to support data related to protein expression, and we attempted to address this issue for the Orai1 antibody and P2Y_12_ and P2Y_13_ receptor antibodies as described below.

In response to reviewer comments, we generated Orai1 CRISPR-knockout iPSC-microglia and further tested the Orai1 Ab (Alomone, Cat# ALM-025, Clone 3F11/D10/B9) for immunostaining in WT and Orai1 KO iPSC-microglial cell line; antibody staining was found to be nonspecific. We therefore agree with the reviewer that, because of poor quality, Orai1 antibody staining does not provide solid and quantifiable evidence for Orai1 expression or function. Accordingly, we have removed Orai1 immunofluorescence staining data (old Figure 1—figure supplement 2E) from the revised manuscript. Instead, we performed Ca^2+^ measurements in the newly generated Orai1 KO iPSC-microglia cell-line, and with a more specific pharmacological inhibitor of Orai channels (Synta66) to unambiguously establish the role of Orai1 in mediating SOCE and in maintaining sustained Ca^2+^ signals by ADP in microglia (new Figure 3—figure supplement 1E and F).

Concerning P2Y expression, we performed control experiments using iPSC microglia treated with siRNA against P2Y_12_ and P2Y_13_ receptors to validate the P2Y_12_ and P2Y_13_ receptor antibodies (P2Y_12_ receptor Ab: Sigma, cat# HPA014518, polyclonal; P2Y_13_ receptor Ab: Alomone, cat# APR-017, polyclonal) for immunostaining. We agree with the reviewer that there is significant non-membrane staining of the cells with the P2Y_12_ receptor antibody making visualization of P2Y_12_ receptors on the membrane problematic. Upon further investigation, the P2Y_13_ receptor antibody was also found to be non-specific, in line with recent reports on the widespread lack of efficacy and specificity of available antibodies to label P2Y13 receptors in ex vivo and in vitro settings including the one used in our study (Alomone, Cat# APR-017; see PMID: 31520551; Suppl. Figure 1). We have therefore removed P2Y_12_ and P2Y_13_ immunofluorescence data from the paper. Instead, we now include flow cytometry data using a different antibody that targets the extracellular domain of the P2Y_12_ receptor (clone 16001E) to show that these receptors are expressed to a greater level on the plasma membrane (PM) of TREM2 KO cells (new Figure 2G). The appropriate isotype control is included in this assay.

2) Important: It is clear that the TREM2 KO cells have significantly lower STIM1 expression, which would be expected to impact SOCE under conditions of physiological agonists stimulation (ADP, ATP). The conclusion based on maximal activation with thapsigargin does not address this issue. What is the impact of reduced STIM1 expression on the Ca signals and motility?

We thank the reviewer for bringing attention to this issue. The volcano plot comparing the transcriptomic expression of STIM1 in WT and TREM2 KO cells shows that STIM1 mRNA expression is below the threshold for what would be considered significant in an RNA seq experiment. In fact, this is the case for all the STIM and Orai isoforms. For RNAsequencing experiments, tens of thousands of genes are tested against the null hypothesis which has led to the use of false discovery rates (FDR) rather than traditional p-values (PMID: 12883005). However, statistical significance based on FDR alone is not enough to determine a meaningful result in RNA-sequencing experiments. Due to highly accurate sequencing, samples may show low variability leading to highly “significant” FDRs with a fold change less than 1. In many cases, this small change in expression would not yield biological differences. For these reasons, we consider only genes that reach both the FDR threshold and the fold change threshold. The data shown here were originally published in (PMID: 33097708) with cutoffs of FDR < 0.05 and -1 < FC < 1.

Unfortunately, the bar-graph in old Figure 1—figure supplement 2D may have given the false impression that differences in STIM1 expression are quite large. We have now remade this bar-graph (Y-axis range from 0 – 1.2) and have also included relative expression of P2Y_12_ and P2Y_13_ receptor transcripts for comparison (new Figure 2—figure supplement 1E). STIM1 mRNA expression is modestly reduced in TREM2 KO cells which is consistent with the modestly reduced maximum functional SOCE response (measured after store-depletion with TG). To address the issue about SOCE under conditions of physiological agonist stimulation, we also measured SOCE in response to store-depletion with ADP; new Figure 3D shows that ADP produces greater store-release and therefore engages SOCE to a greater extent in TREM2 KO cells. Based on this, we conclude that the modestly reduced STIM1 expression in TREM2 KO microglia does not play a significant role in determining the differences in ADP-mediated Ca^2+^ signals between WT and TREM2 KO cells.

3) A Ca^2+^ off/on protocol with ADP (Figure 1) as done for thapsigargin should be performed. In Figure 2B, ADP was applied in the absence of external Ca^2+^, but sadly those recordings did not subsequently replenish Ca^2+^ to gauge Ca^2+^ entry. Low concentrations of Gd3+ (which should not alter Ca^2+^ release) reduce the initial Ca^2+^ release peak, suggesting that there is a contribution of Ca^2+^ entry to this initial response. Also, it would have been helpful to include in Figure 1 Suppl1 the control (non KO cells) in these experiments for reference.

We thank the reviewer for three excellent suggestions (see our replies i, ii, and iii below).

3a (i) To demonstrate SOCE in response to ADP, we now include a Ca^2+^ off/on protocol in the presence or absence of Synta66, a more specific inhibitor of Orai channels than Gd^3+^ or 2-APB. Synta66 significantly inhibited SOCE triggered by TG and ADP in WT and TREM2 KO microglia (Figure 3A, B and Figure 3—figure supplement 1A, B). Additionally, we compared ADP-induced SOCE in WT and TREM2 KO microglia (Figure 3D). The implications of this experiment have been discussed above in Essential revision comment 1 and 2.

3a (ii) We re-examined the issue of whether Ca^2+^ influx contributes to the initial Ca^2+^ peak in two ways. First, we compared the height of the initial Ca^2+^ peak after application of ADP in 1mM Ca^2+^ and Ca^2+^ free buffer (Figure 2—figure supplement 1C); there was no significant difference between the two conditions in either WT or TREM2 KO microglia suggesting that the initial Ca^2+^ peak is driven primarily by store-release. We further tested this by acute addition of Gd^3+^ or 2-APB with ADP without pre-incubation (Figure 3—figure supplement 1C, D). In this instance, we did not find an inhibition of initial Ca^2+^ peak. We speculate that the pre-incubation of cells with Gd^3+^ before addition of ADP in old Figure 1—figure supplement 1 might have non-specifically caused reduction of the initial Ca^2+^ peak.

3a (iii) Given the complex effects of Gd^3+^ and 2-APB on Ca^2+^ signaling in microglia, we did not repeat the Gd^3+^ and 2-APB experiments in WT microglia. Instead, we include new data as outlined above in Figure 3 and Figure 3—figure supplement 1 showing involvement of Orai channels using Synta66 and using the Orai1 KO line (Figure 3—figure supplement 1E, F).

4) The conclusion that SOCE is mediated by Orai1 should be validated with a genetic approach as 2-APB and La3+ are non-specific.

Thank you for the suggestion. As described above, we now include data using a more specific inhibitor (Synta66) and a newly generated Orai1 KO microglial cell line (Figure 3A, B and Figure 3—figure supplement 1A, B, E, F).

5) Conversely, it is not clear if store content is depleted to a greater extent following ADP receptor stimulation in the KO cells. Assess ER calcium content in response to ADP in WT and KO cells or with an ionomycin pulse.

We thank the reviewer for this suggestion. We examined the extent of ER store-depletion in response to ADP by sequentially pulsing the cells first with ADP and then with ionomycin in Ca^2+^ free buffer. The Ca^2+^ release peak in response to ADP was higher in the TREM2 KO cells as expected, and the subsequent ionomycin peak was significantly reduced. These results (Figure 3E and Figure 3—figure supplement 2A, B) indicate greater ER Ca^2+^ store-release by ADP in TREM2 KO cells. As for the suggestion to directly monitor ER calcium content, transfection of iPSC-microglia shifts them from a resting to a highly activated state in which they downregulate their P2Y receptors, as reported (PMID: 17115040, 28602351, 28930663). Thus, transfection of genetically-encoded probes to measure ER Ca^2+^ or to measure IP3 levels is problematic.

6) Figure 3: The WT data in this figure don't include KO cells. How do the TREM2 KOs compare to WT in the scratch wound closure assay? The KO cells seem to be tested, but only in the absence of ADP (Figure 4 Sup 1A). Is there a difference in wound closure after ADP treatment? This point should be evaluated to assess the effects of altered ADP Ca signaling on non-directional cell movement in a physiologically relevant paradigm.

As suggested by the reviewer, we performed the scratch wound assay in the presence of ADP (data now included as Figure 7—figure supplement 1, and described in lines 309-313). Interestingly, while ADP speeds up closure of scratch wounds, we found no differences in the wound closure rates between WT and TREM2 KO microglia. The differences in purinergic signaling are more decisive in shaping chemotaxis to ADP. The effects of TREM2 deletion, and the subsequent effects of increased purinergic signaling on microglial motility appear to depend on the specific physiological context, and this will require further investigation in a follow up study.

7) Terminology: The term "Ca clamp" in the way it has been used here could be misconstrued by readers. Cellular Ca is not clamped. It varies within each cell depending on the activity of the pumps and other clearance mechanisms in the cell. The motility measurements are simply done after extracellular Ca is restored to the cells after thapsigargin treatment, at time points where the cytosolic Ca is lower in 0.2 mM Ca compared to 2 mM Ca. This phrasing (Ca clamp) needs to be changed to eliminate confusion that cellular calcium is "clamped" analogous to voltage-clamp.

We agree with this comment, and have accordingly removed the term “Ca^2+^ clamp” from the manuscript. Instead, we refer to this protocol as a method to investigate the effects of Ca^2+^ elevation that bypasses purinergic receptor activation on microglial motility.

8) Figure 6D TREM2 KOs (but not WT) cells show differences in motility between 0.2 mM and 2 mM extracellular Ca even after thapsigargin treatment with no ADP exposure. This is strong indication that motility differences in TREM2 KO cells are due to more than just alterations in purinergic signaling. (Consistent with lack of additional ADP effects in Figure 6 Supplement 1). What is the role of these intrinsic changes in cell motility in response to cellular Ca elevations in the presence and absence of nucleotides?

We agree with the comment. The question raised has great potential for further work that is beyond the scope of this study. Purinergic stimulation produces a complex downstream response that includes activation of both Ca^2+^-dependent and -independent (Gβ/γ -> PI3K -> PIP3, Ras, cAMP etc) pathways. Experiments in Figure 6 (using a protocol to maintain different levels of cytosolic Ca^2+^ over time using thapsigargin) were done with the goal of isolating the effects of sustained cytosolic Ca^2+^ levels on motility. The key observation is that motility in TREM2 KO microglia responds to changes in cytoplasmic Ca^2+^ levels to a greater extent than in WT cells, suggesting that Ca^2+^ tunes motility differently in WT and TREM2 KO cells. Addition of ADP had no further effect, as the reviewer correctly observes (Figure 6—figure supplement 1). We speculate in the Results section (lines 291-292) that the rise in Ca^2+^ in this assay may override the complex effects of ADP on motility; this may reflect intrinsic differences in Ca^2+^-dependent regulation of motility between WT and TREM2 KO cells. We note that the baseline motility characteristics are similar between WT and TREM2 KO microglia (Figure 5—figure supplement 1A).

9) Figure 6D: The data for WT cells should be directly compared to the KOs at each extracellular Ca concentration to dissect out differences in KO cells at each Ca condition.

Yes, this is shown in Figure 6F: instantaneous speeds of WT and TREM2 KO microglia over a range of different cytosolic Ca^2+^ levels. Additionally, the histogram in Figure 6G compares the percent of cells with instantaneous speeds > 10 µm/min as a function of cytosolic Ca^2+^ in WT and TREM2 KO cells. This comparison reveals a unimodal relationship in TREM2 KO cells with low cell speeds at high and low cytosolic Ca^2+^ levels and higher speeds at intermediate Ca^2+^ levels.

10) The data in Figure 4K should be reanalyzed to compare differences in the absolute levels of branch complexity (# of branches and process length) in the two groups.

We fully agree with the reviewer that the lower baseline branch complexity in the KOs explains the greater fold change in the KOs after ADP application. We also agree that the absolute number and length of branches (normalized to cell number) after ADP treatment is similar between WT and KOs. Based on these comments, we made the following change in the Results section (lines 267-270*),* “Comparison of the absolute number of branches and process length after ADP treatment, as well as the relative fold-increase in these parameters from baseline indicated that process extension is not affected in TREM2 KO microglia”. We note that the absolute number of branches and the process length (normalized to cell number in each imaging field) was calculated and is shown in the left panels of Figure 5L and M. These results are also shown as paired plots comparing the absolute increase in the average number of branches and length of the processes after ADP treatment in the WT and KO groups per imaging field (Figure 5—figure supplement 2A and B, top row).

11) The apparent disconnect between the larger mean square distance travelled in the absence of nucleotides in the KO with no change in the mean speed needs to be clarified. Is this due to cancelling of speed vectors with opposite directions?

We thank the reviewer for bringing attention to this. The MSD vs time plots (Figure 5B) show that TREM2 KO cells travel farther away from the origin than WT cells after ADP treatment. This could either be because (1) KO cells move faster than WT cells or (2) because they change direction less frequently (or travel in straighter paths) than WT cells while moving at similar speed. Comparing mean track speeds and track straightness (Figure 5C) affirms the latter possibility. The mean track speed (Figure 4C-E, Figure 5C, H) is the mean of all instantaneous speeds for a given track. It is calculated without any regard to the direction of cell motility, and therefore does not have negative values. We agree with the reviewer that the velocity vectors (which take into account the direction of cell movement) cancel each other out to a greater degree in the WT cells, which is reflected in lower values of track straightness. Our data suggest that WT cells remain confined to a smaller region of random walk because they turn more frequently. We have observed analogous distinctions in motility patterns between regulatory T (Treg) cells and inflammatory Th17 cells in the spinal cord of mice, with Treg cells executing a back-n-forth motion with similar mean speeds as Th17 cells, and being confined to smaller regions of the cord while Th17 cells traverse larger regions because they change directions less frequently (PMID:32732436). We now include a sentence in the Discussion section (lines 349-350) to indicate that velocity vectors cancel each other out to a greater extent in WT cells. For clarity, we added a new paragraph in the methods section (lines 922-937) detailing how the motility parameters were calculated.

Reviewer #1 (Recommendations for the authors):1) Figure 3: The WT data in this figure don't include KO cells. How do the KOs compare to the WT in the scratch wound closure assay? The KO cells seem to be tested, but only in the absence of ADP (Figure 4 Sup 1A). Is there a difference in wound closure after ADP treatment? This point should be evaluated to assess the effects of altered ADP Ca signaling on non-directional cell movement in a physiologically relevant paradigm.

We thank the reviewer for this suggestion. We have addressed this under Essential revision comment no. 5.

2) Figure 4: This reviewer is not convinced that reduced turning accounts for the similar speed in the KO despite the larger mean square distance travelled. If turning is reduced and the mean square distance is doubled in the KO, then effectively, the average distance travelled over time (speed) should be greater in the KO than in the WT. The logic of this inference and how it explains the data (similar average speed but much greater root mean square displacement in the KO) needs to be better clarified. Since the mean square distance is so much larger, it suggests that the speed vectors with opposite directions are cancelling out in the KO cells to yield the same average velocity (which could theoretically even occur from increased, not decreased turning).

We thank the reviewer for bringing attention to this issue. We have addressed this under Essential revision comment no. 10.

3) Figure 4K: It seems that the baseline branch complexity is low to start with in the KOs, hence the fold change from baseline seems larger. From the images shown, there appears to be no difference following ADP treatment between the two groups. This data should be reanalyzed to compare differences in the absolute levels of branch complexity (# of branches and process length) in the two groups.

We thank the reviewer for bringing attention to this issue. We have addressed this under Essential revision comment no. 9.

4) The use of the term Ca clamp is misleading in the way it has been used here. Cellular Ca is, in fact, not clamped, but changes within each cell depending on the activity of pumps and other calcium clearance mechanisms. The motility measurements are simply done after extracellular Ca is restored to cells after thapsigargin treatment, at time points where the cytosolic Ca is lower in 0.2 mM Cacompared to 2 mM Ca. This phrasing (Ca clamp) needs to be changed to eliminate confusion that cellular Ca levels are "clamped" to defined values analogous to voltage clamp.

We agree with this comment, and have accordingly removed the term “Ca^2+^ clamp” from the manuscript (Essential revision comment no. 6).

5) Figure 6D: It is noteworthy that TREM2 KOs, but not WT cells, show differences in motility between 0.2 mM and 2 mM extracellular Ca even after thapsigargin treatment with no ADP exposure. This is strong suggestion that motility differences in TREM2 KO cells are due to more than just alterations in purinergic signaling. In fact, this inference is consistent by the lack of additional effects of adding ADP in Figure 6 Supplement 1. These intrinsic changes in cell motility in response to cellular Ca elevations need further clarification.

We have addressed this under Essential revision comment no. 7.

6) Figure 6D: Related to this figure, the data for WT cells should be directly compared to the KOs at each extracellular Ca concentration to dissect out differences in KO cells at each Ca condition.

We have addressed this under Essential revision comment no. 8.

Other:– Stim1 is expressed at different levels between the WT and TREM2 KO. It has a significant p-value (Figure 1 Sup2D). This decrease could affect multiple aspects of microglia including cell motility in the presence of physiologically relevant agonists. This needs to be addressed.

We thank the reviewer for this comment. This point was made by other reviewers as well and needs better clarification. We have addressed this under Essential revision comment no. 2.

– A technical point related to Figure 4 is that figures 4C and 4G should be labelled "velocity" rather than speed, as it seems the only way in which the mean squared distance can be twice as large in the KO but not the velocity if the speed vectors with opposite directions are cancelling out in the KO.

These graphs (current Figure 4C-E, current Figure 5C, H) plot the mean track speed, which represents the mean of all instantaneous speeds over the total time of tracking. The instantaneous speed is calculated at each time point for a given track as the scalar equivalent to object velocity, without taking into account directionality (hence no negative values). We agree that MSD is lower in WTs; this is because WT cells turn more frequently (reflected as lower track straightness compared with TREM2 KOs), leading to greater cancellation of velocity vectors. For clarification, we now include definition of the different motility parameters in lines 922-937 of the methods section.

– The dose of ADP used in Figure 7 is extremely high. 100ng/mL (which is 234uM by my calculation). By contrast, Figure 4 used 2.5uM. The 100 fold higher concentration in the chemotaxis assay needs to be justified. Further, it should be determined whether the ADP used diffuses into both chambers might be expected. This super high dose of ADP may have non-physiological effects that could explain the seemingly differing phenotypes in Figures 4 and 7.

The reviewer’s calculation of concentration is incorrect. 100 ng/ml of ADP is equivalent to a concentration of 234 nM, not 234 µM. In the methods section on the chemotaxis assay (line 962), we now also report the concentration of ADP in nM. This dose of ADP is not super high.

– Figure 5 is a great set of experiments to show that the Salsa6f does not affect end points of the study, but may be better as a supplementary figure rather than a main figure. If the paper is about the TREM2 KO versus WT then the main figures should stick with looking at differences between the two with maybe a sub figure to show validation.

We thank the reviewer for this suggestion and are happy to comply. Because we ended up using the Salsa6f expressing lines in several additional experiments to confirm key findings in the study, the Salsa6f validation data are now presented in Figure 1 figure supplement 1.

Reviewer #2 (Recommendations for the authors):– A more rigorous approach could be used to document the presence or absence of compensatory changes at the protein protein level of other Ca^2+^ signaling proteins (Figure 1 Suppl2). Preferably Westerns should be performed for STIM1 and STIM2, Orai1 and IP3R 1/2/3, for which there are reliable Abs. Although IF as performed for Orai1 is acceptable, the authors do not show positive and negative controls and a thorough validation of this Orai1Ab in IF.

We thank the reviewer for these comments and suggestions. We have addressed this under Essential revision cmment no. 1.

– Along the same lines as comment above, Staining for P2YR12 appears internal. Has the Ab been validated? Can authors perform a WB on P2YR12 and P2RY13 in addition?

We have addressed this under Essential revision comment no. 1.

– Did the authors check on whether the expression of SECRA is altered at the protein level? Did the authors perform CRAC channel recordings in Control and TREM2 KO cells?

We have not looked at the expression of SERCA at the protein level. However, we further examined our published RNAseq data on WT and TREM2 KO iPSC-microglia (PMID: 33097708) and compared the relative read counts of PMCA and SERCA isoforms that are known to be expressed in WT and TREM2 KO microglia and found no significant differences. Data for the relative expression of relevant Ca^2+^ signaling molecules (STIM1, Orai1, IP3R2, SERCA2 and 3, PMCA1) are now shown in Figure 2—figure supplement 1D and E. We note that among these, only P2Y_12_ and P2Y_13_ receptor transcripts showed significant fold change in expression between WT and TREM2 KO cells, which prompted us to focus our efforts in that direction. Additionally, we found that Ca^2+^ clearance rate after SOCE is similar in WT and TREM2 KO cells (Figure 3—figure supplement 2E, F). Based on this, we find it unlikely that the higher sustained Ca^2+^ level in TREM2 KO cells is due to differences in Ca^2+^ clearance mechanisms, including Ca^2+^ pump activity. We did not attempt to record Icrac in iPSC microglia.

– It is quite surprising that the authors did not opt to use the Ca^2+^ off/on protocol with ADP (Figure 1) as they have done with thapsigargin. In Figure 2B, ADP was applied in the absence of external Ca^2+^, but sadly those recordings did not subsequently replenish Ca^2+^ to gauge Ca^2+^ entry. One can rationalize the result of 2-APB (Figure 1 Suppl1), a non specific channel blocker that also inhibits IP3Rs. However, the fact that low concentrations of Gd3+ (which should not alter Ca^2+^ release) reduce the initial Ca^2+^ release peak, suggests that there is a contribution of Ca^2+^ entry to this initial response. Also, it would have been helpful to include in Figure 1 Suppl1 the control (non KO cells) in these experiments for reference.

We thank the reviewer for these comments and suggestions. We have accordingly performed new experiments and address this under Essential revision comment no. 3a.

Additional thoughts to the attention of the authors:– The authors have speculated on the fact that increased cytosolic Ca^2+^ might lead to a "spill" outside the restricted Ca^2+^ nanodomain, thus disrupting polarity. Do the authors have any evidence of altered location of leading-edge vs trailing edge proteins or focal adhesion proteins in TREM2 KO cells?

No, we did not investigate the interesting question regarding the polarity of membrane proteins or focal adhesion proteins in TREM2 KO cells. Ca^2+^ signals play a role in generating cell polarity and regulating membrane protrusion and retraction in some migratory cell-types (PMID: 25977921). Local Ca^2+^ pulses have been detected at the leading edge of migrating fibroblasts in response to PDGF gradients (PMID: 19118385), and in migrating sheets of endothelial cells in response to a scratch wound (PMID: 24463606). More recently, localized Ca^2+^ signaling has been proposed to correlate with process extension in murine microglia based on intravital imaging, though it’s role in chemotaxis was not explored (PMID: 32716294). We have now rephrased the relevant sentence in the Discussion section (lines 360-363) to make it clear that we are speculating on the possible mechanism by which excessive Ca^2+^ signaling can impede gradient sensing. Reflecting the uncertainties, we have removed Figure 8 considering it may be overly speculative.

– Have the authors performed RNAseq on WT control and TREM2 KO cells? If so, does pathway analysis shows changes in the Cell motility genes between these two groups? This might offer some interesting clues to pursue in future studies.

The authors have performed and previously published RNA-seq on WT and TREM2 KO cells (Figure 1 in PMID: 33097708). Using gene ontology analysis, the authors did find differences in motility including “regulations of natural killer cell chemotaxis”, “Positive regulation of cell migration”, and “regulation of smooth muscle cell migration”. Even in the more selective gene ontology of genes which change in opposite directions after TREM2 KO and TREM2 antibody stimulation, we find “positive regulation of leukocyte chemotaxis” as one of the most significant gene family suggesting differences in motility. These gene lists included P2RY_12_ and P2RY_13_ receptors and were part of what stimulated our work presented here. However, some other chemotactic molecules including CCL2 and CCL3 are also included in this reciprocal gene list and warrant further study.

Reviewer #3 (Recommendations for the authors):This study by Jairaman et al. describes how iPSC-derived microglia exhibit exaggerated cytosolic Ca^2+^ responses to ADP stimulation in TREM2KO cells, and that this leads to a defect in turning behaviour and hence no directed migration to a chemotactic signal.Overall, the experiments are well conducted, carefully controlled and the findings are new and exciting. The authors nicely dissect out the underlying molecular basis for the exaggerated Ca^2+^ responses to ADP and then extend their findings to cell movement and directed migration. Given the substantial body of evidence linking microglia to the pathogenesis of Alzheimer's disease, and the role for TREM2, the work by Jairaman et al. is of translational significance. As an aside, the introduction of the calcium-sensitive reporter Salsa6F is a welcome new tool in the arsenal for recording cytosolic calcium. Overall, this is an elegant, novel and important study. Nevertheless, I have a few comments/suggestions.

We thank the reviewer for the positive and insightful comments. We have done new experiments to address specific suggestions as outlined below.

1. The authors argue that the increased Ca^2+^ plateau to ADP in TREM2KO cells is due to enhanced Ca^2+^ release from the stores ie greater store depletion. Evidence is presented that maximal SOCE is not compromised in TREM2KO cells, that store content under resting conditions is unaffected, InsP3R activities are similar. But the authors do not demonstrate that store content falls more following ADP receptor stimulation in the KO cells. This could be shown by directly measuring ER calcium to ADP in WT and KO cells or by applying ADP in Ca-free solution and then assessing store content with an ionomycin pulse. The ionomycin response should be smaller in the TREM2KO cells, after ADP exposure.

We thank the reviewer for suggesting the ionomycin pulse experiment. We have accordingly performed new experiments, and address this under Essential revision comment# 3b.

2. The authors suggest that an increase in P2Y12 and P2Y13 receptor expression in the KO cells accounts for the increased Ca^2+^ release from the ER. However, the increase in P2Y12 protein levels is modest at best, and the increase for P2y13 is less than 2-fold. The authors' point would be strengthened by showing InsP3 levels are increased in the KO cells compared with WT ones, for the same dose of ADP. Single cell InsP3 probes are available. Alternatively, a population measurement could be carried out.

While we agree that the suggested experiment with InsP3 probe would further bolster our conclusion, we have noted in essential revision comment no. 4 that transfection of iPSC-microglia shifts them from a resting to a highly activated state leading to downregulation of P2Y_12_ receptors, which confounds our ability to accurately compare P2Y receptor activity. Although the antibodies against P2Y_12_ and P2Y_13_ receptors were found to be nonspecific (see also essential revision comment No. 1), we now include flow cytometry data showing increased PM expression of P2Y_12_ receptor in live TREM2 KO cells (Figure 2G); these data complement the functional assays and transcriptomic data (Figure 2—figure supplement 1E) showing >2 fold increase in P2Y_12_ and P2Y_13_ receptor RNA expression in TREM2 KO microglia. These differences are likely to be biologically relevant for downstream signaling, given the positive feedback mechanisms and signal amplification associated with GPCR signaling (PMID: 29074251). Increased protein expression of P2Y_12_ and P2Y_13_ receptors has also been reported in *Trem2^-/-^* mice and in a mouse AD model, and we now include this point in the Introduction section of the manuscript (line 87-89).

3. The evidence for an involvement of SOCE is based on two rather non-specific inhibitors (2-APB and Gd3+). P2YRs, like other GPCRs, activate TRPC proteins and these Ca^2+^-permeable channels could contribute to the Ca^2+^ plateau. The authors should therefore knock down Orai1 or at least use more selective inhibitors such as Synta66 or the GSK compound.

We thank the reviewer for this suggestion and have performed additional pharmacological experiments and CRISPRbased Orai1 KO to address this point in Figure 3 and Figure 3—figure supplement 1. The results confirm involvement of Orai channels in SOCE activated by store-depletion with TG and ADP.

4. The authors should include some controls to show the P2Y antibodies are indeed specific. For example, they could use a cell line that lacks P2Y12/13. In Figure 2I, J, what happens to the ADP response when both P2Y inhibitors are present at the same time? It would be important to show that the combination of P2Y12 and P2Y13 suppress ADP responses fully.

Per reviewer suggestion, we examined Ca^2+^ responses to ADP in the presence of both P2RY_12_ and P2Y_13_ inhibitors. This combination inhibited the ADP Ca^2+^ response completely in both WT and TREM2 KO microglia (Figure 2G).

5. The authors should add a justification for the concentrations of agonists they have used (ADP, ATP, UTP). A dose-response curve is included for ADP but a comment on the doses selected would be helpful.

Extracellular concentration of purinergic signals range from hundreds of nanomolar to μM levels, and are shaped by a variety of factors including baseline secretion, extent of local tissue damage and the prevalence of ectonucleotidases that cleave purinergic ligands (PMID: 18302942). Studies in microglial field have often used tens of μM ADP (PMID: 17115040, 11245682) to study the biology of P2Y_12_ receptors, but microglial cells experience a range of concentrations depending on the pathophysiological context, and depending on distance from the injury site. We now include a sentence in the Introduction (line 54) about purinergic concentrations in the brain.

6. It is nicely shown that Ca^2+^ influx in TREM2 KO cells leads to microglia motility and process extension to a greater extent than WT cells. This is attributed to the increased SOCE. If so, then one might expect raising external Ca^2+^ in WT cells to have the same effect. Is this the case?

We have not examined effects of increasing the extracellular Ca^2+^ concentration in WTs to test whether that results in a higher magnitude of cell speed and process extension (compared to TREM2 KOs). Our data in Figure 5 and Figure 5 figure supplement 1 shows that Ca^2+^ influx pathways are required for optimal cell motility and process dynamics in iPSC microglia in general (based on greater increases in cell track speeds, process length and branching in response to ADP in 1 mM Ca^2+^ compared with 0 mM Ca^2+^ extracellular solution for both WT and TREM2 KO microglia). We should also note that the effects of TREM2 deletion on process branching/ lengthening is not as drastic as its effects on cell motility. We have addressed this particular point in response to comments from reviewer No. 1 under Essential revision, comment no. 9.

7. Is anything known mechanistically how the exaggerated Ca^2+^ signal leads to a chemotaxis defect? The authors' data would suggest that chemotaxis might have a bell-shaped dependence on cytosolic Ca^2+^, with too little or too much impeding migration towards a cue.

The fundamental issue in chemotaxis is how shallow gradients of a chemokine translates to steep polarization of signaling proteins within the cell. This is generally thought to occur downstream of the chemokine receptor and upstream of actin cytoskeleton (PMID: 33990789), but the specific role of Ca^2+^ signals in this process remains poorly understood. Please also see our earlier response (comment No. 5 by reviewer# 2) on how excessive cytosolic Ca^2+^ may disrupt cell polarity. The histogram in Figure 6G comparing the percent of fast-moving cells as a function of cytosolic Ca^2+^ further indicate that there may indeed be a bell-shaped effect with regard to effects of Ca^2+^ on instantaneous cell speeds in TREM2 KO cells. This is reflected in the chemotaxis assay in Figure 7B and C in which reducing ADP signaling in TREM2 KO cells rescues the defect in chemotaxis. It is possible that high cytosolic Ca^2+^ serves as a temporary STOP signal in microglia similar to its effects on T cells. We note this point in the Discussion (lines 386-388) and speculate that TREM2 KO cells may be more subject to this effect with ADP, given the higher expression of P2RY_12_ and P2Y_13_ receptors.

8. Do the authors know whether the effects of SOCE are mediated through a local Ca^2+^ rise or via a global cytosolic Ca elevation?

We have not looked at role of local vs. global Ca^2+^ signals in regulating microglial cell motility in the current study but hope to explore this aspect of Ca^2+^ signaling as part of a follow up study.

9. Perhaps I missed something but Figure 1 S2(D) seems to show a significant decrease in STIM1 levels in the KO cells.

We thank the reviewer for this comment. This point was made by other reviewers as well and needs clarification. We have addressed this under Essential revision comment no. 2.

10. The authors present SOCE as the peak signal remaining after 5 minutes. It is not clear whether the value has been subtracted from the pre-stimulation levels. In some graphs, the signal after 5 minutes looks the same as the resting level, but the bar charts show higher values.

We assume the reviewer is referring to the traces and bar-graphs in current Figure 1B, D, E and F showing both peak Ca^2+^ response and Ca^2+^ at 5 minutes. The data-points (single cell values) in the bar-graphs are all baseline subtracted, as described in the legend. The Y-axis is now labelled “Fluo-4/Fura-Red Ratio (baseline subtracted)” to make this point clear. We also note that the bar-graph summary includes data from multiple experiments, while the average traces are from a single imaging run.